



# Reviews and syntheses: Bacterial bioluminescence – ecology and impact in the biological carbon pump

Lisa Tanet[1], Séverine Martini[1], Laurie Casalot[1], Christian Tamburini[1]

[1]Aix Marseille Univ., Université de Toulon, CNRS, IRD, MIO UM 110, 13288, Marseille, France

*Correspondence:* Christian Tamburini (christian.tamburini@mio.osupytheas.fr)

**Abstract.** Around thirty species of marine bacteria can emit light, a critical characteristic in the oceanic environment where the major part is deprived of sunlight. In this article, we first review current knowledge on bioluminescent bacteria symbiosis in light organs. Then, focusing on gut-associated bacteria, we highlight that recent works, based on omics methods, confirm previous claims about the prominence of bioluminescent bacterial species in fish guts. Such host-symbiont relationships are relatively well established and represent important knowledge in the bioluminescence field. However, the consequences of bioluminescent bacteria continuously released from light organ and through the digestive tracts to the seawater have been barely taken into account at the ecological and biogeochemical level. For too long neglected, we propose to consider the role of bioluminescent bacteria, and to reconsider the biological carbon pump taking into account the bioluminescence effect ("bioluminescence shunt hypothesis"). Indeed, it has been shown that marine snow and fecal pellets are often luminous due to microbial colonization, which makes them a visual target. These luminous particles seem preferentially consumed by organisms of higher trophic levels in comparison to non-luminous ones. As a consequence, the sinking rate of consumed particles could be either increased (due to repackaging) or reduced (due to sloppy feeding or coprophagy/coprorhexy) which can imply a major impact on global biological carbon fluxes. Finally, we propose a strategy, at a worldwide scale, relying on recently developed instrumentation and methodological tools to quantify the impact of bioluminescent bacteria in the biological carbon pump.

## 1 Introduction

Darkness constitutes the main feature of the Ocean. Indeed, the dark ocean represents more than 94 % of the Earth's habitable volume (Haddock et al., 2017). Moreover, the surface waters are also in dim light or darkness during nighttime. Organisms living in the dark ocean biome are disconnected from the planet primary source of light. They must adapt to a continuous decrease in sunlight reaching total darkness beyond a few hundred meters. Hence, it is not surprising that 76 % of marine pelagic meso- and macro-organisms are bioluminescent from the surface to the deep sea, without variability over



depth and that bioluminescence is a major ecological function in interactions (Martini and Haddock, 2017). Bioluminescent

species are found in most phyla from fish to bacteria (Haddock et al., 2010; Widder, 2010). Amongst marine light-emitting

organisms, luminous bacteria are the most abundant and are widely distributed. Most of the 30 currently known bacterial

luminous species are heterotrophic, copiotrophic and facultatively anaerobic. Endowed with important motility and

chemotactic abilities, luminous bacteria are able to colonize a large variety of habitats (as symbionts in light organs and guts,

free-living in seawater or attached to particles) (e.g. (Dunlap and Kita-tsukamoto, 2006) and references therein). Bacterial

bioluminescence is energetically costly, and its benefices are understood in its symbiotic form. On another note, bacterial

bioluminescence in its free or attached forms is still to be explained. A barely investigated pathway is the bioluminescence

contribution into the biological carbon pump.

The biological and physical (solubility) carbon pumps are the main drivers of the downward transfer of carbon and play a

central role in the sequestration of carbon dioxide (Boyd et al., 2019; Buesseler and Lampitt, 2008; Dall'Olmo et al., 2016).

The biological carbon pump is defined as the process through which photosynthetic organisms convert $CO_2$ to organic

carbon, as well as the export and fate of the organic carbon sinking from the surface layer to the dark ocean by different

pathways (Siegel et al., 2016). Sinking particles (greater than 0.5 mm of diameter) known as marine snow are a combination

of phytodetritus, living and dead organisms, fecal pellets (from zooplankton and fish). Marine snow, rich in carbon and

nutrients, and their surrounding solute plumes are hotspots of microbial activity in aquatic systems (Alldredge et al., 1990;

Alldredge and Silver, 1988; DeLong et al., 1993). Marine snow is also consumed by zooplankton, and fecal pellets are a

food source through coprophagy. When leaving the epipelagic zone and sinking to depth, organic particles would be utilized

by microbial decomposition and fish/zooplankton consumption, both considered as responsible for a large part of the

variation in the efficiency of the biological carbon pump (De La Rocha and Passow, 2007). Recently, fragmentation

(potentially due to biological processes in the mesopelagic waters) has also been shown to be the primary process controlling

the sequestration of sinking organic carbon, accounting for $49 \pm 22\%$ of the observed flux loss (Briggs et al., 2020).

In this review, we will summarize the current knowledge on bioluminescent bacteria based on former and recent literature.

First, we describe symbiotic bioluminescent bacteria in light organs of fish or squids, its importance and controls. Then, we

present enteric-association occurrences and their potential role for the host. One of the consequences of these symbioses, in

both light organs and guts, is a massive quantity of bioluminescent bacteria daily dispersed in the ocean. Based on this

statement, we claim and demonstrate that bioluminescent bacteria have an ecological and a biogeochemical importance in

the biological carbon pump, catalyzing and amplifying the involved processes. We propose a synthetic representation of the

bioluminescence shunt of the biological carbon pump and a future strategy to establish and quantify the impact of

bioluminescence **(Figure 1).**



## 2 Symbiotic bioluminescent bacteria in light organs

In Eukaryotes, light emission has two distinct origins: intrinsic or symbiotic (Haddock et al., 2010; Nealson, 1979). Intrinsic luminescence is caused by chemicals produced by the organism itself. Most bioluminescent organisms are self-luminescent and have specialized luminous cells called photophores (Herring, 1977). Some animals, however, are capable of luminescence using symbiotic luminous bacteria housed in elaborate and specialized organs.

## 2.1 Discovery, importance, distribution and functions of light-organ symbiosis

In the late 1880s, Raphaël Dubois was among the first to suggest bacteria to be responsible for the light emitted by some animals (Harvey, 1957). In the beginning of the twentieth century, Balthazar Osorio (1912) provided clear and convincing evidences of such symbiosis, when luminescent bacteria were described in high density within dedicated fish gland, called the light organ (Hickling, 1926). Since then, luminous bacterial symbiosis has been the subject of interest among the scientific community working on bioluminescence, to such an extent that, by the mid-twentieth century, luminescence of many organisms was thought to have bacterial origin. However, some of these assessments have been refuted later (Herring, 1977).

From a species level perspective, bioluminescence ability is shared by about 8 % of all known fishes (Paitio et al., 2016). Amongst luminous fishes, bacterial luminescence is the rule for almost half of them (48 %) (Davis et al., 2016). To date, symbiotic bacteria are recognized as responsible for the luminescence of ray-finned fishes and some squids (Davis et al., 2016; Haygood, 1993; Lindgren et al., 2012). Although forms of symbiotic luminescence have been suggested for some shark species or pyrosomes (Dunlap and Urbanczyk, 2013; Leisman et al., 1980), no evidence of luminous bacteria have been found so far (Claes and Mallefet, 2009; Renwart et al., 2014; Widder, 2002) and a recent study has definitely rejected a bacterial origin in the velvet belly lanternshark (Duchatelet et al., 2019). Concerning luminous squids, intrinsic bioluminescence is more common, and symbiotic light organs are known in two families (Sepiolidae and Loliginidae) (Lindgren et al., 2012; Nishiguchi et al., 2004).

Symbiotic luminescence seems more common in benthic or coastal environments for fish and squid as well (Haygood, 1993; Lindgren et al., 2012; Paitio et al., 2016). Shallow-water fishes with luminous bacterial symbionts include flashlight fishes (Anomalopidae), ponyfishes (Leiognathidae) and pinecone fishes (Monocentridae) (Davis et al., 2016; Morin, 1983). For deep-sea fishes, anglerfishes (Ceratiodei) and cods (Moridae) are among the common examples of luminous-bacteria hosts.

In general, the origin of light production, intrinsic or symbiotic, is the same within a host clade. However, while all other Apogonidae exhibit intrinsic light, the *Siphamia* species host luminous bacteria (Paitio et al., 2016). Another exception concerns a genus of anglerfishes, *Lynophryne*, which possesses both systems of light production, having intrinsic luminescent barbel in addition to a symbiotic luminous esca (Hansen and Herring, 1977). To date, presence of this dual



system in an organism is unique among all known luminous animals (Pietsch et al., 2007). Bacterial and intrinsic light
organs are predominantly intern and in ventral location (Paitio et al., 2016; Wilson and Hastings, 2013). Due to the position
of some internal light organs, localized within the coelomic cavity, therefore away from the taxonomic examination process,
the luminescence ability of some fishes has remained unrecognized for a long time (Haneda and Johnson, 1962).

Fish and squid with bacterial light organs likely use the emitted light to conceal themselves by counterillumination,
obliterating their silhouette, therefore avoiding dusk-active piscivorous predators (Jones and Nishiguchi, 2004; McFall-Ngai
and Morin, 1991). Less common but more striking, some organisms found in the families Monocentridae, Anomalopidae and
numerous deep-sea anglerfishes belonging to the suborder Ceratoidei, exhibit light organs colonized by bacteria (Haygood,
1993). These light organs are thought to be predominantly used to illuminate nearby environment or attract prey or mates.

## 2.2 Symbiont selection and colonization of the light organ

01 Like most symbiotic bacterial associations with animals, luminous bacteria are acquired from the surrounding environment
02 by individuals, independently of their ancestry (i.e. horizontally transmitted) (McFall-Ngai, 2014).

03 Knowledge of the mechanisms involved in the selection and the establishment of bacterial symbionts have considerably
04 improved in last decades. Harvest of the luminous symbionts from the bacterioplankton is driven by microbial recognition
05 and molecular dialog (Kremer et al., 2013; Nyholm et al., 2000; Nyholm and McFall-Ngai, 2004; Pankey et al., 2017;
06 Schwartzman and Ruby, 2016; Visick and Ruby, 2006). Bacterial colonization of host tissues induces the morphogenesis
07 process of the light organ and appears to signal its further development and maturation (McFall-Ngai and Ruby, 1991;
08 Montgomery and McFall-Ngai, 1998). The luminescence feature is essential for a correct morphogenesis process of the light
09 organ and symbiont persistence inside (McFall-Ngai et al., 2012; Visick et al., 2000). One of the best-documented symbioses
is the association of *Aliivibrio fischeri* with the bobtail squid *Euprymna scolopes* (Nyholm and McFall-Ngai, 2004; Ruby,
1996). Through the easy independently cultivation of both partners in laboratory, this symbiosis has become a perfect model
for studying the process of bacterial colonization into the light organ, and understanding bacteria–animal interactions,
broadly speaking (Mandel and Dunn, 2016; McFall-Ngai, 2014). *E. scolopes* squid is able to reject non-luminous strains of
*A. fischeri* (Bose et al., 2008; Koch et al., 2014), suggesting that the host possesses the capability of detecting (at a molecular
or physiological level) if its symbiont is bioluminescent or not (Miyashiro and Ruby, 2012; Peyer et al., 2014; Tong et al.,
2009). Additionally, a genetic distinction between strains of the same bacterial species, such as the presence of two operons
containing the light-emission-involved genes (Ast et al., 2007), is sufficient to avoid a successful colonization of the light
organ in a given host (Urbanczyk et al., 2012).

Although it was previously reported that symbionts from light organs were all members of the genus *Photobacterium*
(Nealson and Hastings, 1979), we now know through taxonomic reclassifications and the rise of acquired knowledge, that
they are not restricted to this clade. To date, 11 species are known to be involved in light-organ symbioses **(Table 1)**. In a
light organ, the bacterial population is most of the time monospecific (Dunlap and Urbanczyk, 2013; Ruby, 1996). Thus,





organisms with light organ perform bioluminescent-bacteria batch culture as microbiologists try to do. Interestingly enough,

it is one of the rare bacterial cultures done *in situ* by marine organisms. Although light organs are generally colonized by a

unique species, existence of genetically distinct strains have been reported for some *E. scolopes* (Wollenberg and Ruby,

2009). Moreover, in the light organ of certain squid and fish, two species of luminous bacteria can co-occur. Indeed, light

organ of some *Sepiola* spp. are colonized by a mixed population of *A. fischeri* and *A. logei* (Fidopiastis et al., 1998). The *P.*

*mandapamensis* and *P. leiognathi* species are also co-symbionts of some Perciformes fish (Kaeding et al., 2007). In the same

vein, some loliginid squids have been found to harbor a consortium of several luminous species in their light organ,

including at least *P. angustum*, *P. leiognathi* and *V. harveyi* (Guerrero-Ferreira et al., 2013).

The host-symbiont specificity appears consistent at the species level **(see Table 1)**. However, this is not true at the host

family taxonomic level (Dunlap et al., 2007). Moreover, multiple unrelated host species are colonized by the same symbiont

species. These symbiont strains present no clear phylogenetic divergence between themselves, revealing no evidence of

codivergence between symbiont and host. Such a lack of strict symbiont/host specificity and codivergence in luminescence

symbiosis may be due to the environmental acquisition of luminous bacteria at each new generation rather than a parental

transmission which could favor higher genetic speciation (Dunlap et al., 2007).

Considering that fish and squid housing luminous bacteria are never found without symbionts in nature, the symbiosis

appears obligatory for hosts (Haygood, 1993). In contrast, most symbiotic bacteria are viable outside the light organ, and

thus are considered as facultatively symbiotic. These facultative symbiotic bacteria are readily culturable under laboratory

conditions, outside the host light organ. Exceptions have been highlighted for the luminous symbionts of two groups of fish,

the flashlight fish (family Anomalopidae) and the deep-sea anglerfish (suborder Ceratiodei) (Dunlap and Kita-tsukamoto,

2006; Haygood and Distel, 1993). Indeed, despite the fact that the bacterial origin of the light was proved by microscopic

observation and that genes from luminous bacteria were amplified (Haygood and Distel, 1993), bacterial cultivation has been

yet unsuccessful. Thanks to the emergence of genome sequencing, complete genome of these symbionts has been reported in

the last years. Analyses revealed a genome reduction in size by about 50 % and 80 % for anglerfish and flashlight-fish

symbionts respectively, compared to facultative luminous symbionts or free-living relatives (Hendry et al., 2014, 2018).

Genome reduction is a common trait shared by bacteria involved in obligatory symbiosis (Moran et al., 2009) and explains

the inability of these symbionts to grow in laboratory cultures. Flashlight-fish and anglerfish symbionts appear to be

obligatory dependent on their hosts for growth, as some metabolic capacities (e.g. genes necessary for amino acid synthesis)

are absent in the genome.

**2.3 Light organs are under well-established controls**

Although light organs can differ in form, size or location according to the host **(see Table 1)**, some structural and functional

features are common for all of them. The light organ is a separate and highly evolved entity. Luminous bacteria are densely

packed within tubules which communicate to the exterior of the light organ (Haygood, 1993; Nealson, 1979). The host

provides nutrients and oxygen to the tubules through a highly vascularized system (Tebo et al., 1979). Bioluminescent





bacteria, which are not directly affected by mechanical stimulation, emit light continuously in the light organ, as they do in

laboratory cultures (Nealson and Hastings, 1979). However, the light intensity varies over time. As for self-luminescent fish,

bacterial light organs have evolved with multitude of adaptations of tissue, to serve as reflectors, diffusers, screens, and light-

conducting channels (Haygood, 1993; Munk et al., 1998). Such anatomical features assist in directing and enhancing light

output (Sparks et al., 2005). In addition, the host can control the light diffusion through different mechanisms, which may be

external lids, chromatophores, organ rotation, filters, occlusion with a shutter, or muscle contraction (Hansen and Herring,

1977; Herring, 1977; Johnson and Rosenblatt, 1988). As example, for counterillumination, controlling the intensity of light

output gives the host a better camouflage, adapting its silhouette to environmental changes in light (Jones and Nishiguchi,

2004; McFall-Ngai and Morin, 1991). For intra-species communication, it permits to produce sudden flashes or specific

signal/rhythm of light (e.g. schooling behavior (Gruber et al., 2019)).

In squid-*vibrio* symbiosis, bacterial luminescence genes are regulated with quorum-sensing system, a cell-density-dependent

process. When the cell density reaches a certain level, autoinducers responsible for triggering the synthesis of the genes

involved in light emission are accumulated in sufficient amount, and light is emitted (Nealson et al., 1970; Verma and

Miyashiro, 2013). Variation of light emission is closely linked to the concentration of one component involved in the

bacterial light reaction, which could be host controlled. Interestingly, *A. fischeri* produces a higher level of luminescence

within the light organ than in laboratory cultures, despite a similarly-high cell density (Boettcher and Ruby, 1990). Hence,

Verma and Miyashiro (2013), suggested that the light organ environment offers specific conditions such as the levels of

oxygen, iron, or phosphate, to enhance bacterial light emission.

Within the light organ, luminous symbionts reach a very high density which reduced the oxygen availability, essential for the

light reaction. Such oxygen limitation leads to a decrease in the specific luminescence activity (Boettcher et al., 1996).

Bacterial population inside the light organ is regulated by the host, by coupling the restriction of the growth rate and the

expulsion of symbionts. Growth repression is thought to reduce the energetic cost of the symbiosis to the host (Haygood et

al., 1984; Ruby and Asato, 1993; Tebo et al., 1979). Additionally, the cell number of symbionts is regulated by the daily

expulsion of most of the bacterial population, followed by a period of regrowth of the remaining symbionts. This periodical

released is highly correlated with the diel pattern of the host behavior. For example, in squid-vibrio symbiosis, the host

expels 95 % of the luminous symbionts in the surrounding environment at dawn, the beginning of its inactive phase. The

remaining 5 % of *A. fischeri* grow through the day and the highest concentration is reached at the end of afternoon, at the

nocturnal active phase of the squid (Nyholm and McFall-Ngai, 2004; Ruby, 1996). For all symbioses, luminous bacteria in

excess, densely packed inside tubules communicating with the exterior of the light organ, are regularly expelled (Haygood,

1993). Regular expulsion of symbionts maintains favorable conditions in the light organ for the bacterial population, but it

also seeds the environment with luminous symbionts for colonization of the next host generation. The consequence is a

release of a huge quantity of bioluminescent bacteria in the seawater inducing a major contribution to the ocean microbiome.

To make it more concrete and provide an order of magnitude, two examples are proposed thereafter. Using laboratory

experiments on different fishes (Monocentridae, Anomalopidae), Haygood et al. (1984) estimated a release between $10^7$ to





$10^9$ bioluminescent bacterial cells per day and per individual. Another study on the Hawaiian bobtail squid (*E. scolopes*) has

estimated that the squid expels about 5 x $10^8$ bioluminescent bacterial cells per day and per individual (Lee and Ruby, 1994).

These discharges lead to a regular luminous-bacteria enrichment of the areas inhabited by this organism.

Depending on the anatomical location of the light organ **(see Table 1)**, luminous symbionts are released directly into the

surrounding seawater or through the digestive tract (Haygood, 1993; Nealson and Hastings, 1979). An enteric lifestyle has

indeed been suggested for the luminous bacteria (Ruby and Morin, 1979; Nealson, 1979).

## 3 Enteric associations

The gastrointestinal (GI) tract of an animal is a very complex and dynamic microbial ecosystem (Nayak, 2010). Current

knowledge and concepts on GI microbiota derive from studies on humans or other terrestrial mammals. In contrast, GI

00   ecosystems of marine inhabitants have yet received little attention, and studies focused on farmed fish or commercially

01   important species of fish. Whether aerobes or anaerobes are the main group in the microbiota in fish intestines is still

02   discussed (Romero et al., 2014). For marine fish, the dominant members seem to be facultative anaerobes (Wang et al.,

03   2018). Considering that most of the bioluminescent bacteria are facultatively anaerobes (Ramesh et al., 1990; Reichelt and

04   Baumann, 1973), it is not surprising to find them in gut niches.

05

### 3.1 Occurrence in marine-fish guts

07   Although luminescence of dead fish was a well-known phenomenon, one of the first mentions of the presence of luminescent

08   bacteria in fish slime and intestinal contents is only from the beginning of the 1930's (Stewart, 1932). Since then, the high

09   occurrence of luminous bacteria in fish intestines has been reported in many studies (Baguet and Marechal, 1976; Barak and

Ulitzur, 1980; Liston, 1957; Makemson and Hermosa, 1999; O'Brien and Sizemore, 1979; Ramesh and Venugopalan, 1988;

Reichelt and Baumann, 1973; Ruby and Morin, 1979). Most of hosts with internal light organ release luminous bacteria into

the digestive tract (Haygood, 1993; Nealson and Hastings, 1979), and thus may largely contribute to their abundance in

luminous fish intestines. However, one interesting case concerns a leiognathid fish, which internal light organ is colonized

by *P. leiognathi*. Although its light organ is directly connected to its digestive tract (Dunlap, 1984), the luminous enteric

population was not dominated by *P. leiognathi* (33 %), but by *V. harveyi* (67 %) (Ramesh et al., 1990). Actually, many

fishes without light organ also harbor luminescent bacteria in their gut (Makemson and Hermosa, 1999), which clearly

demonstrates existence of other sources for enteric luminous bacteria.

Through the gut-content analysis of 109 fish species from the Gulf of Oman, Makemson and Hermosa (1999) showed that

the relative proportion of the occurring culturable luminous bacteria was strongly variable. While some fish guts harbor more

than 80 % of luminous bacteria, some others have between 20-50 %, and a minority have none detected, with a substantial



intra and inter-species fish variability. As other authors, Makemson and Hermosa (1999) highlighted *V. harveyi* and *P.*

*phosphoreum* as the dominant luminous species found in fish guts (Reichelt and Baumann, 1973; O'Brien and Sizemore,

1979; Ramesh and Venugopalan, 1988). Interestingly, a high proportion of luminescent bacteria (>70 %) has been found in

the gut of an Atlantic halibut recently fed, while an individual male in spawning condition, that had not been eating recently,

had a flora dominated by non-luminescent microorganisms (Verner-Jeffreys et al., 2003). This result underlines the link

between food ingestion and abundance of luminous bacteria and suggests that they do not persist within the halibut gut once

the feces are eliminated. This also suggests that luminous bacteria are then released with the feces in the water column.

Seasonal variations have been observed in both luminous bacterial density (Liston, 1957; Ramesh and Venugopalan, 1988),

and predominant species (Bazhenov et al., 2019). Such variability is not surprising since it is inferred to the structure and

composition of the gut microbiota of fish which is influenced by a series of factors, including (i) host factors (e. g genetics,

gender, weight, age, immunity, trophic level), (ii) environmental factors such as water, diet, and surrounding environment,

(iii) microbial factors (e.g. adhesion capacity, enzymes and metabolic capacity), (iv) and individual variations and day-to-

133   day fluctuations (Nayak, 2010; Sullam et al., 2012; Wang et al., 2018). Hence, contrasting results can be found in the

literature:  for example, a dominance of the *Clostridium* (a non-luminous clade) is commonly associated with herbivorous

fishes (Clements et al., 2009), while *Vibrio* and *Photobacterium* (which are clades with luminous representatives) are the

dominant genera in carnivorous fish diet (Egerton et al., 2018). In contrast, Makemson and Hermosa (1999) have reported a

slightly higher proportion of culturable luminous bacteria in herbivore fish compared to carnivore. They also emphasized the

higher incidence of luminescent bacteria in pelagic than in reef-associated fish, as well as filter-feeder-fish guts contain more

luminous bacteria compared to other feeding type (e.g. predator). For bigger fishes, a potential introduction source of

luminous bacteria into gut could be the ingestion of smaller preys bearing bacterial light organ. For all organisms, enteric

luminous bacteria may be transferred to the gut bacterial community of their predators.

It should be emphasized that investigations on microbial communities of fish have long been limited by the use of culture-

dependent methods (Austin, 2006; Romero et al., 2014). We now know that only a small proportion of microorganisms can

be cultivated under laboratory conditions (Amann et al., 1995). Moreover, the fish-gut microbiota has been reported to be

particularly of low cultivability, with less than 0.1 % of the total microbial community cultivable (Zhou et al., 2014),

although the level of cultivability may be taxon dependent (Ward et al., 2009). Today, advanced molecular techniques offer a

wide variety of culture-independent methods, such as Next-Generation Sequencing (NGS), for analyzing fish microbiota

(Tarnecki et al., 2017). As a consequence, it is appropriate to investigate if luminous microbiota constitute a significant

portion of the total gut microbiota of fish as it has been suggested in previous works mentioned above, or if this trend was

distorted by the use of culture-dependent methods.

Several studies using gene sequencing based on 16S rRNA to characterize the gut microbiome of fish have reported the

genus *Photobacterium* as the most abundant in the guts of salmon and trout (Bagi et al., 2018; Givens et al., 2015; Michl et

al., 2019; Riiser et al., 2018), shark (Michl et al., 2019) and Atlantic cod (Bagi et al., 2018; Givens et al., 2015; Michl et al.,

2019; Riiser et al., 2018). Other studies reported the presence of *Photobacterium* spp. in the gut of hydrothermal shrimp



(Durand et al., 2009), and, seasonally variable, in the gut of Norway lobster (Meziti et al., 2010). However, because not all

*Photobacterium* spp. have luminescence ability, it is important to be able to resolve dominant OTU at the species level,

which, most of the time, is not possible with a 16S rRNA barcoding sequencing approach. The emergence of multi-gene

approaches offers more detailed insights into the taxonomic diversity of these communities (i.e. species level). Thus, using

metagenomic shotgun sequencing, two independent and recent works on wild Atlantic cods also concluded of the

*Photobacterium* spp. domination and have been able to go deeper into the taxonomic identification. Le Doujet et al. (2019)

demonstrated that *Photobacterium* genus represents 78 % of all present genera and identified the *P. phosphoreum* clade as

the most abundant *Photobacterium* lineage. According to Riiser et al. (2019), the luminous species *P. kishitanii* constitutes

over 26 % of the Vibrionales community, which is the dominant clade, and the authors underlined the presence of the

functional *lux* genes. Therefore, recent metagenomic studies seem to confirm the trend of a high occurrence of luminous

bacteria in fish intestines.

## 3.2 Are enteric luminous bacteria playing a specific role for the host?

From their presence in GI tract, the enteric bacteria may gain rich-nutrient accessibility. In reply, GI microbial communities

may play critical roles on host health, development and nutrition (Romero et al., 2014; Wang et al., 2018). A clear

understanding of the role that the specific gut microbiota plays is still lacking. It has been highlighted that components of the

bacterial microflora are associated with several functions, such as epithelial renewal, amino-acid production, complex-

molecule degradation, or inhibitory-compound secretion, that protect host against bacterial pathogen colonization (Austin,

2006; Wang et al., 2018). However, little is known about a possible role of enteric luminous bacteria on the host physiology.

A rare item is that some luminous bacteria, and particularly *Photobacterium* spp., may contribute to the digestion of complex

molecules, like for example, being involved in chitin degradation (Ramesh and Venugopalan, 1989; Spencer, 1961).

Pathogen processes related to bioluminescent bacteria are regularly investigated and reviewed (Austin and Zhang, 2006;

Dunlap and Urbanczyk, 2013; Fidopiastis et al., 1999; Nelson et al., 2007; Ramesh and Mohanraju, 2019; Wang et al.,

2015). Many luminous bacteria can act as opportunistic pathogens, and particularly on marine crustaceans, by entering the

body of animals through lesions on its surface. However, such opportunistic pathogen behavior is not specific to luminous

bacteria, but their presence is probably highlighted due to the visible light emitted (Dunlap and Urbanczyk, 2013).

Based on the increase in light emission observed on dead marine animals, Wada et al. (1995) argue that, at the death of the

host, enteric luminous bacteria may have an important saprophytic lifestyle. On dead organisms, luminous bacteria present in

the gut of the host could initiate rapid propagation and decomposition of the host body, and result in the formation of

luminous debris in the marine environment. For marine vertebrates, luminous strains of *Photobacterium* spp.,

psychrotolerant and histamine producing, are regularly described as the major spoilage organisms on fish caught and stored

(Barak and Ulitzur, 1980; Bjornsdottir-Butler et al., 2016; Dalgaard et al., 1997; Figge et al., 2014; Macé et al., 2013). In





contrast to dead organisms, on living vertebrate specimens, infection by luminous bacteria rarely occurs (Dunlap and Urbanczyk, 2013).

## 4 Luminous bacteria and the biological carbon pump

As previously discussed, light organs and guts act as a source for luminous-bacteria persistence in the oceans. Therefore, luminous bacteria are widespread in the ocean. They can be found as free-living forms or attached to particles (Nealson and Hastings, 1979; Ramesh and Mohanraju, 2019; Ruby et al., 1980).

### 4.1 Bioluminescent bacteria in the water column

Qualitative and quantitative studies showed that the luminous bacteria are dynamic over time and space. Seasonal variations have been identified, both in abundance and predominant species (O'Brien and Sizemore, 1979; Ruby and Nealson, 1978; Yetinson and Shilo, 1979). A wide variability has been observed in species repartition over depth and between geographic areas (DeLuca, 2006; Gentile et al., 2009; Nealson and Hastings, 1979; Ramaiah and Chandramohan, 1992; Ruby et al., 1980). Horizontal, vertical and seasonal variations were most of the time presumed to reflect physiological preference, and particularly temperature or salinity sensitivity (Orndorff and Colwell, 1980; Ramesh et al., 1990; Ruby and Nealson, 1978; Shilo and Yetinson, 1979; Yetinson and Shilo, 1979). Some works mentioned that symbiotic niches, such as light organs and enteric tracts, may serve to inoculate the planktonic population (Nealson et al., 1984; Nealson and Hastings, 1979; Ramesh et al., 1990; Ruby et al., 1980). To our knowledge, very few studies focused intensively on the contribution of species-specific symbiotic associations on the occurrence and distribution of luminous bacteria in the surrounding water. Amongst these rare studies, Lee and Ruby (1994) reported that the abundance of *A. fischeri*, the luminous symbiont of the Hawaiian squid *E. scolopes* was 24 to 30 times higher, in both water column and sediments, in areas inhabited by the squids than in similar locations where squids were not observed.

Bioluminescent bacteria also seem to be the cause of the spectacular and still largely unexplained events, so-called milky seas (Lapota et al., 1988; Nealson and Hastings, 2006). Milky seas are characterized by an unusual brightness on the ocean surface and extend over such a large area that the light emitted is detectable from space (Miller et al., 2005). The light-emission pattern of milky seas is continuous and homogeneous, which is consistent with light emission from bacteria and easily distinguished from blooms of dinoflagellates.

### 4.2 Bioluminescent bacteria attached to particles

Outside of spatially restricted niches, as light organ or gut environments, role of the dispersed luminous cells in marine environment was matter of debate and it was thus mentioned that non-symbiotic bacteria may have no ecological





significance (Hastings and Greenberg, 1999; Nealson and Hastings, 1979). However, Herren et al. (2004) suggested that luminous bacteria are more attached to particles than free-living, which was confirmed by Al Ali et al. (2010). Many bacteria, including bioluminescent bacteria (Ruby and Asato, 1993; Zhang et al., 2016), can develop swimming behavior to colonize the sinking organic material, therefore reaching a cell density 100 to 10,000 times higher than in the water column (up to $10^8$ to $10^9$ cells mL$^{-1}$) (e.g. Ploug and Grossart, 2000).

Bacteria that glow on particles can attract macro-organisms. After being ingested, they will find a more favorable environment to live and grow in their gut (Andrews et al., 1984; Ruby and Morin, 1979). Actually, this is the preferred current hypothesis that supports a positive selection related to the dispersion and propagation of the bacteria. Indeed, luminous bacteria growing on particulate matter could produce enough light to be visible by other organisms. For bacterial species with light production under cell-density control (i.e. under quorum-sensing regulation), the high cell concentration reached on particles can allow the sufficient accumulation of the autoinducers, and thus the emission of light for attracting predators. For species which light production is not subject to cell-density control (i.e. not under quorum-sensing regulation) (Tanet et al., 2019), to be able to produce light at very low cell concentration could give them an advantage for being prior eaten. Continuously glowing bioluminescent emissions are thought to attract predators (Nealson and Hastings, 1979). In the water column, the glowing bacteria aggregated on particles would lead to the detection, attraction, ingestion and decomposition of particles by larger organisms. Grazers would consume luminous matter at a higher rate than invisible particles. Being consumed and ending up into the gut, bacteria would benefit of a more suitable environment regarding the growth conditions and the nutrient accessibility. In open ocean, and particularly in deep regions, where sparse nutrient supply prevails, rich-nutrient gut niches of the surrounding animals could appear as an oasis of life for bacteria. This dispersion hypothesis has also been strongly consolidated by field data where bacterial bioluminescence was observed in freshly excreted fecal pellets and in materials collected from sediment traps (Andrews et al., 1984), as well as by laboratory experiments where glowing zooplankton were preferentially ingested by fishes (Zarubin et al., 2012).

The copiotrophic type of luminous bacteria is another point supporting their particle-attached lifestyle. Bacterial population colonizing nutrient-rich environments (e.g. floating carcass, marine snow, fecal pellets or the gut tract of a marine eukaryote) are defined as copiotrophs, by opposition to the oligotrophs which are members of free-living microbial populations (Lauro et al., 2009). All luminous marine bacteria from *Vibrio* and *Photobacterium* spp. possess two chromosomes in their genome (Boyd et al., 2015; Zhang et al., 2016), with a high copy number of rRNA operons. Such genomic features, as a large genome size and multiple rRNA operons, are considered as an adaptation for a copiotrophic lifestyle (Klappenbach et al., 2000; Lauro et al., 2009). Copiotrophs are thought to have strong adaptability skills, permitting them to survive long enough between two nutrient-rich environments (Yooseph et al., 2010).

Fish guts could also act as an enrichment vessel for the growth of luminous bacteria, and thus enhance their propagation (Nealson and Hastings, 1979; Ramesh and Venugopalan, 1988). When expelled with feces, enteric luminous bacteria can be easily isolated from the fresh fecal material. This fecal luminescence increased in intensity over a matter of hours, proving that luminous bacteria survived the digestive process and can proliferate on such organic material (Ruby and Morin, 1979).



Henceforth, fish feces appear to be an important source of viable luminous bacteria in the marine environment and could

affect both the distribution and the species composition of luminous populations. The luminescence of fecal particles has

been reported numerous times and was always associated to luminous bacteria, due to the observation of continuous light

emission or direct isolation (Andrews et al., 1984; Ramesh et al., 1990; Raymond and DeVries, 1976; Ruby and Morin,

1979; Zarubin et al., 2012).

In comparison with free-living luminous bacteria, few studies have focused on bioluminescence of marine snow and fecal

pellets. Yet, observations on materials collected from sediment traps revealed light emission in 70 % of all samples, with two

distinct patterns of light kinetics, probably due to the presence of different luminescent organisms (Andrews et al., 1984).

Surface-sample (above 60 m depth) analyses reported that more than 90 % of the luminous-aggregate samples exhibited

bacterial luminescence (Orzech and Nealson, 1984). Another study (between 2 and 17 m depth) also reported a large part of

luminous marine snow, but more likely due to dinoflagellates (Herren et al., 2004).

## 4.3 Bioluminescent bacteria in the sediments

Information relative to luminous bacteria in sediment is also limited. It is known than bioluminescent bacteria can be isolated

from sediment samples (Ramesh et al., 1990), but rare data exist about their distribution or abundance. In some sediment

samples, occurrence of luminous bacteria among total heterotrophic bacteria could reach up to 70 %, with seasonal

variations (Ramesh et al., 1989), although less pronounced than in water column (O'Brien and Sizemore, 1979). Main

sources of luminous bacteria in sediments are likely the glowing sinking marine snow, and benthic or demersal host

harboring symbiotic light organ with regular discharges.

More recently, sediment resuspension events (Durrieu de Madron et al., 2017) were correlated with newly formed deep-

water events and deep-sea bioluminescent events recorded in the NW Mediterranean Sea (Martini et al., 2014; Tamburini et

al., 2013a). Since the presence of active luminous bacteria has been demonstrated on the site (Martini et al., 2016), it has

been hypothesized that resuspended luminescent bacteria present in sediment can be part of these luminescence events

(Durrieu de Madron et al., 2017). Additionally, dense water formation, conveying particulate organic matter, could further

increase luminous bacteria proliferation and activity (Tamburini et al., 2013a).

## 4.4 How do bioluminescent bacteria impact the biological carbon pump?

Based on the ecological versatility of the bacterial bioluminescence reviewed above, we propose to reconsider the classical

view of the fate of organic matter in the oceans. **Figure 1** represents the guideline of the bioluminescence shunt hypothesis

of the biological carbon pump.

Bioluminescent bacterial emissions are continuous over time and such characteristic is thought to attract predators. Indeed,

the light color from bioluminescence contrasts well against the dim or dark background of the ocean depths. In the





bathypelagic zone (1000-4000 m), where no daylight remains, bioluminescent emissions are considered as the major visual

stimulus (Warrant and Locket, 2004; Widder, 2002). For such reason, symbiotic associations have been selected as an

advantage for hosts (fish or squid) in light organs. Luminous bacterial symbionts are successively acquired by juveniles and

released into the seawater to control population concentration **(Figure 1, step 1)**. As indicated previously, the released of

bioluminescent bacteria from light organs and fecal pellets can represent an unbelievable quantity of bioluminescent bacteria

in the water column.

Recent studies underlined the very-well-adapted fish vision to the detection and location of point-source bioluminescence

(Busserolles and Marshall, 2017; Mark et al., 2018; Musilova et al., 2019; Paitio et al., 2016; Warrant and Locket, 2004).

Although less intensively documented than fishes, crustacean (copepods, amphipods, isopods…) visual system is also

reported to have sensitivity shift to bluer wavelength, which aids their bioluminescence detection (Cohen and Forward, 2002;

Frank et al., 2012; Marshall et al., 1999; Nishida et al., 2002). In laboratory experiments, Land et al. (1995) demonstrated

that amphipods where attracted to a blue-light-emitting diode. Unfortunately, and despite these statements, rare studies have

investigated the effect of bioluminescence on the ingestion rates of predators **(Figure 1, step 2)**. To our knowledge, the only

one known is from Zarubin et al. (2012), who experimentally measured 8-times-higher ingestion rate of glowing (due to

ingestion of bioluminescent bacteria) zooplankton by fishes, compared to non-luminous zooplankton. Moreover, they

demonstrated the attraction of zooplankton by luminous bacteria.

Glowing bacteria have been observed attached to particles of organic matter, marine snow and fecal pellets (**Figure 1**, from

00  symbionts in guts in **step 1** and through predation in **step 2**) sinking into the deep ocean. Thus, while sinking into the deep,

01  these glowing bacteria living on organic carbon particles (marine snow, fecal pellets…) would lead to the detection,

02  attraction, ingestion and decomposition of particles by larger organisms. Consumers would ingest luminous matter at a

03  higher rate than invisible particles and consequently will augment luminous-microorganism dispersion by fecal-pellet

04  excretion. Bioluminescent sinking material should accelerate the consumption of organic matter by attracting grazing

05  organisms. Interestingly, bacteria associated with animal guts are thought to be particularly adapted to high-hydrostatic

06  pressure (Deming et al., 1981; Ohwada et al., 1980; ZoBell and Morita, 1957). Indeed, certain bioluminescent bacteria resist

07  to high hydrostatic pressure (Brown et al., 1942), and some of them have a higher growth rate and emit more light than at

08  atmospheric pressure (Martini et al., 2013). Such piezotolerance, or piezophile lifestyle, is undoubtedly an advantage for

09  luminous bacteria attached to particles that are exposed to pressure variations during the sinking-particles fluxes (Tamburini

et al., 2013b). The addition of these bioluminescent tags on particles has two indirect impacts (**Figure 1, steps 2 & 3**). First,

due to aggregate fragmentation by sloppy feeding and coprorhexy, fast-sinking particles are transformed into slow-sinking or

suspended particles. Fragmentation has been shown to be the primary process controlling the sequestration of sinking

organic carbon (Briggs et al., 2020). The second possibility is that organic matter ingestion leads to aggregation by

repackaging, and the excreted pellets of higher density, are fast-sinking particles. Filter-feeder plankton, without visual

detection and food selection by light, will also passively contribute to such aggregation or fragmentation of particles. For

these organisms, bioluminescence can even have a negative effect since they can be identified by the luminous material





filtered. Additionally, the consumption of organic material colonized by bioluminescent bacteria increases their dispersal rate

provided by migrating zooplankton, and even more so by actively swimming fish, following the conveyor-belt hypothesis

(Grossart et al., 2010) **(Figure 1, step 4)**. This dispersion due to the expelling of luminous feces is several orders of

magnitude greater than that of water-borne free bacteria.

Sediment resuspension is another process implying the consumption of luminous bacteria by higher trophic levels **(Figure 1,**

**step 5)**. This potentially re-inseminates bacteria into the bioluminescence loop through the consumption by epi-benthic

organisms.

Considering this bioluminescence shunt hypothesis, all the processes described above show that bioluminescence can be

view as a catalyst in the biological gravitational carbon pump (Boyd et al., 2019), by either increasing the carbon

sequestration into the deep ocean, or by slowing down the sinking rate of particles and consequently increasing their

degradation and the remineralization rate. Bioluminescence and especially luminous bacteria may therefore influence the

export and sequestration of biogenic carbon in the deep oceans. A better quantification of these processes and impacts in the

biological carbon pump are a requirement in future studies.

## 5 Past and future instrumentation for bioluminescence assays

### 5.1 Previous sampling methods to describe diversity and abundance of luminous bacteria

In the existing literature, to estimate the diversity and the distribution of bioluminescent bacteria, studies were based on a

restricted number of sampling methods and instruments. These methods focused either on environmental samplings where

bacteria are present, or on organisms with associated bacteria.

First, vertical samplings in the water column were performed using sterile-bag samplers (Ruby et al., 1980), or later, using

Niskin bottles (mounted on rosette profilers) (Al Ali et al., 2010; Gentile et al., 2009; Kita-Tsukamoto et al., 2006; Martini et

al., 2016; Yetinson and Shilo, 1979). This approach is commonly set up in oceanography but rely on relatively small

volumes of water (up to 20L). Furthermore, it does not fully capture the heterogeneity of the ecosystem since it provides one

discreet sample over restricted time and space. Other instruments dedicated to the acquisition of sediment sampling are

the multiple-core samplers, deployed onto the seafloor (Kita-Tsukamoto et al., 2006). For particulate organic carbon and

fecal pellets, in order to describe the diversity of associated luminous bacteria, sediment traps have been occasionally

deployed from the surface down to the deep ocean (Andrews et al., 1984). Using them, fresh luminous material has been

collected between 30 to 1900 m depth down.

To study the presence of luminous symbionts in guts and light organs larger organisms are caught. The most common way to

catch deep-sea animals is the deployment of trawls and more generally nets. They are well adapted to sample squid

(Zamborsky and Nishiguchi, 2011) or fishes, like the anglerfish (Freed et al., 2019). One particularity of these methods is

that the sampling covers a large section of the water column and pulled everything into one catch with a limited precision



about depth layers. SCUBA diving is another method to gently select these large animals (Zamborsky and Nishiguchi, 2011).

It has also been used to catch fecal pellets and sinking particles (Orzech and Nealson, 1984). Obviously, SCUBA diving has

a strong depth limitation (generally above 50 m depth). It can be more efficient at night for some migrating species and has a

restricted sampling size of organisms and number of samples carried back to the ship.

Once environmental samples or material from organism's light organs have been acquired, the objective is either to describe

the taxonomy and diversity of luminous bacteria, or to quantify them. To do so, earlier studies have filtered seawater samples

through a polycarbonate filter with a pore size of 0.2 µm to retain bacteria. The filter is then placed with the bacterial side up

on growth medium in petri dishes (Kita-Tsukamoto et al., 2006; Ruby et al., 1980). For symbiotic bacteria, light organ or

guts are aseptically dissected shortly after death, and the content is homogenized before culture or microscopic observations

(Dunlap, 1984). After hours of incubation, the total colony forming units is observed; the luminous colonies can, then, be

enumerated and selected for taxonomic investigation.

Further investigations of symbiotic associations, in relation to surrounding environment, would require a reliable taxonomy

of luminous bacteria and robust knowledge on species-specific symbiotic associations. As an example, *Photobacterium*

*phosphoreum* was thought to be the specific symbiont of light organ of numerous deep-sea fish (Hendrie et al., 1970; Ruby

et al., 1980; Ruby and Morin, 1978), before a phylogenetic analysis showed distinct evolutionary lineages in the *P.*

*phosphoreum* clade according to the colonized habitat. This resolution revealed that all the *P. phosphoreum* symbionts

isolated from light organ should actually be identified as *P. kishitanii* (Ast and Dunlap, 2005).

## 5.2 Future strategy to quantify the role of bioluminescence in the biological carbon cycle

Since these first investigations on luminous bacteria in symbioses or in the environment, there has been a huge improvement

in technology and molecular-biology techniques. To better evaluate the role of bioluminescence and luminous bacteria in the

biological carbon pump further studies have to follow an efficient strategy. Such strategy will focus on quantifying this

functional trait and how it impacts the transfer of organic carbon between trophic levels, as well as its sequestration into the

deep ocean. This approach can be divided into several key points 1) the assessment of the global importance of

bioluminescence in the oceans, 2) the pursue of investigations about the quantification and diversity of luminous bacteria and

their variability between ecosystems (free-living in the water column, on sinking particles and fecal pellets, or in sediments),

3) the quantification of luminous bacterial release into the surrounding environment and the potential impact of vertical

migration, and 4) the quantification of the transfer rate of bacteria attached on glowing particles into zooplankton and the

quantification of the effects on organic matter decomposition, sinking rate and fluxes, in comparison to non-glowing

particles. In this review, future perspectives to allow major advances on these specific key points are proposed based on

technologies recently developed.



### 5.2.1 The assessment of the global importance of bioluminescence in the oceans

In order to establish the global importance of light emitted by organisms, which include glowing bacteria, quantitative surveys are needed at large spatial scales including geographical variability and depth. Current existing fixed platforms (including observatories), oceanographic vessels, remotely-operated and autonomous underwater vehicles (AUV), and gliders have considerably increased our knowledge of marine ecosystems and their spatial variability. For temporal scales, in the last decades, the multiplication of long-term observatories and ongoing European *in situ*-observing-infrastructure initiatives, such as the Fixed-point Open-Ocean Observatories (FixO3), the European Multidisciplinary Seafloor Observatory (EMSO), the European Research Infrastructure, or the ARGO International Program (EuroArgo) (Favali and Beranzoli, 2009; Le Reste et al., 2016) have increased global-ocean observations at long time scales (more than 10 years) and high sampling frequency. To quantitatively record bioluminescence emissions, some instruments are commercially available, or have been adapted from existing sensors. Bathyphotometers, a system pumping water into a closed chamber and measuring the emission of light by a photomultiplier, are the most commonly used (Herren et al., 2005), and have already been implemented on AUV (Berge et al., 2012; Messié et al., 2019; Moline et al., 2009) and other vertical profilers (Cronin et al., 2016). Other approaches have been developed unexpectedly from astrophysics telescopes using photomultipliers with a very high sensitivity to photons embedded into optical modules. These instruments have been proved to be efficient to detect bioluminescence in deep-sea environments and over long-time surveys (Aguzzi et al., 2017; Martini et al., 2014; Tamburini et al., 2013a). Another example of quantitative records of photon counts is the equipment of bio-samplers, such as elephant seals, with a small, autonomous tag recording environmental light and bioluminescence. These tags have been shown to be a great improvement in highlighting ecological functions such as predator/prey relationships and could inform on the role of bioluminescent prey for seals (Vacquié-Garcia et al., 2012). The technological development of high sensitivity cameras has opened another path for bioluminescence exploration. Low light cameras have been used to record *in situ* light patterns (Maxmen, 2018; Phillips et al., 2016) and implemented on remotely operated vehicles for direct *in situ* observations of sinking particles, or marine luminescent creatures.

Theoretically, both bacterial, glowing continuously, as well as eukaryotic light, emitted as flashes, could be detected. All of these instruments, with the capability to record surrounding or mechanically stimulated light, have been extensively developed or adapted within the last 10 years. Their future implementation on multiple observatories and vehicles will definitely increase our knowledge on the global importance of bioluminescence in the oceans. Long-time surveys could elucidate extreme observed events, such as, the bacterial abundance in water-mass movements and sediment resuspension (Durrieu de Madron et al., 2017) or the frequency of milky seas (Lapota et al., 1988; Miller et al., 2005) due to luminous bacteria. Over space, profilers will provide information about the role of bioluminescence in vertical nychthemeral migrations. However, the future challenge is that the deployment of these instruments has to be done in parallel with data analysis. Acquisition of quantitative signal will induce the discrimination of different groups of organisms including



bacteria, and, consequently, will require the development of strong statistical methods in signal analysis (Messié et al., 2019).

To go deeper than *in situ* quantitative observations, samplings are necessary in various ecosystems including marine snow, water column, sediments, as well as light organs of fishes and squids.

### 5.2.2 The pursue of investigations about the quantification and diversity of luminous bacteria and their variability between ecosystems (free-living in the water column, on sinking particles and fecal pellets, or in sediments)

Marine snow potentially glows due to luminous micro-organisms colonizing these habitats (bacteria, eukaryotes), but there are only few studies, based on limited numbers of samples that have quantified luminous bacteria on marine snow in the dark ocean. A first step is to establish the extent of glowing particles over depth, to assess if this is a common or marginal phenomenon. This can be done either by direct observation of light or by describing the biodiversity associated to these particles. Particles are difficult to sample due to their fragility. However, vehicles such as remotely operated vehicles are able to collect particles of marine snow at specific depth using suction samplers and bring them back to the surface into biological collectors. Sediment samplers, potentially implemented on benthic rover, are other instruments used to sample marine snow, fecal pellets and particles. This is already a common tool deployed during oceanographic cruises but samples from sediment traps are generally dedicated to biogeochemistry analyses which involve fixing their content using reagents. To assess the activity of luminous bacteria, it will only require keeping this material fresh without fixing reagent in order to observe the light emission. Glowing aggregates can be observed by using low light cameras and the light measured by photomultipliers. After observations, these samples can be used for multiple biogeochemical analyses including bacterial taxonomic diversity and abundance.

### 5.2.3 The quantification of luminous bacteria in the environment and the potential impact of vertical migration

The analysis of water and sediment samplings can considerably be improved by omics methods to pursue investigations of bacterial taxonomic diversity and functions and assess their variability between different ecosystems (including sediments, marine snow, and water column).

Advances in Next Generation Sequencing (NGS) methods open new opportunities to describe the structure of communities and the part of luminous bacterial strains present in environmental samples. These methods are an opportunity to sequence bacterial species even if it is not cultivable, which has been one major limitation of traditional methods. In order to efficiently describe bioluminescent or non-bioluminescent bacteria, the description at the species level is a strong requirement. As an example, *Vibrio* are important contributors to particulate organic carbon fluxes that have been observed at abyssal depths in the Pacific Ocean (Preston et al., 2019, Boeuf et al., 2019). A better characterization at species or functional level should highlight the luminous potential related to the presence of such organisms, even at low abundance.



Metabarcoding and transcriptomic could also be used on particles and fecal pellets sampled over depth to describe the
biogeography of luminous bacteria.

One track for further investigations is to take advantage of large sampling efforts to sample at a global scale made with
oceanographic cruises such as TARA Ocean, Tara Polar circle circumpolar expeditions (Pesant et al., 2015) or
MALASPINA (Duarte, 2015). These expeditions have established a protocol to provide consistent methodology on the
analysis of micro-organism biodiversity. The data available could give some new inputs on the variability of luminous
bacteria over ecosystems around the globe.

**5.2.4 The quantification of the transfer rate of bacteria attached on glowing particles to consumers and the effect on**
**organic matter decomposition, sinking rate and fluxes, in comparison to non-glowing particles**

One main lock to evaluate the importance of bioluminescence in the biological carbon pump is to quantify the transfer rate of
organic carbon between trophic levels. Few studies related the preferential consumption of luminous bacteria by zooplankton
(copepods in Nishida et al., 2002) or fish (Zarubin et al., 2012). In the laboratory, investigations on processes influencing
consumption rates of zooplankton on glowing particles can be performed to define the parameters inducing these higher
attraction rates. Future studies based on the experimental protocol described by Zarubin et al. (2012) could be improved by
including other zooplankton species of importance in the biological carbon pump and multiple bacterial species. In a dark
room, under controlled conditions (close to *in situ*) the attraction rate of glowing (fresh or infected by luminous bacteria) and
non-glowing aggregates can be tested on zooplankton (copepods, mysids) as well as higher trophic levels (small fish). The
effect of temperature, bacteria species, abundance/diversity of zooplankton communities, glowing/non-glowing particles,
light intensity, hydrostatic pressure and other variables can be tested on particles attraction behavior. One main improvement
is the capability of low-light cameras to record associated behaviors under the laboratory experiments.

**6 Conclusion**

Light organ and gut of marine animals act as reservoirs for the abundance and persistence of luminous bacteria in the ocean.
Additionally to light organs and gut niches, bioluminescent bacteria colonize particles of organic-matter, making them
glowing. Taking into account the powerful attraction of luminescence on fish and zooplankton consumption, luminous
bacteria may therefore influence, in different ways, the export and sequestration of biogenic carbon in oceans. Finally, a
multi-instrumented strategy will definitely increase knowledge on bioluminescence and the role of luminous bacteria in the
biological carbon pump. This strategy can be set up based on both traditional methods and recently developed technology
and is promising in the near future.



## Author contributions:

LT and CT proposed the idea. LT provided the first version of the review. The following authors were in charge of the initial draft of the corresponding sections: LT: luminous bacteria in light organs and guts, spatial distribution of luminous bacteria, SM: role of luminous bacteria into the biological carbon pump and future strategy. LC and CT supervised the work. LT, SM, LC and CT wrote, reviewed and edited the final review.

## Competing interests:

The authors declare that they have no conflict of interest.

## Acknowledgements

LT was supported by a doctoral grant "Région Sud" and TANGRAM Architectes agency. We gratefully acknowledge support from CNRS (Project EC2CO "HEMERA"). The project leading to this publication has received funding from European FEDER Fund under project 1166-39417. We thank H.P Grossart and J. Mallefet for providing helpful comments on an earlier version of this review.

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





**Figure and Table captions.**

Figure 1: Bioluminescence shunt in the biological carbon pump in the ocean. Luminous bacteria in light-organ symbioses are successively acquired by host (squid, fish) from the seawater while they are juveniles, then regularly released into the ocean. Depending on the light-organ position, luminous bacteria are released from their guts into fecal pellets or directly into the seawater (step 1). Motile luminous bacteria colonize organic matter sinking along the water column. Bioluminescent bacteria inseminating fecal pellets and particles influence zooplankton consumption rates. Such visual markers increase detection ("bait hypothesis"), attraction and finally predation by upper trophic levels (step 2). In the mesopelagic, zooplankton and their predators feed on sinking luminous particles and fecal pellets, which either form aggregates (repackaging) of faster sinking rates or fragment organic matter (due to sloppy feeding) with slower sinking rates (step 3). Filter feeders also aggregate sinking organic matter without particular visual detection and selection of luminous matter. Diel (and seasonal) vertical migrators feeding on luminous food, metabolize and release glowing fecal pellets from the surface to the mesopelagic zone (step 4). It implies bioluminescent bacteria dispersion at large spatial scales, for zooplankton or even some fish actively swimming on long distances. Luminous bacteria attached on particles sink down to the seafloor, sediment can be resuspended by oceanographic physical conditions (step 5) and consumed by epi-benthic organisms. Instruments area: (a) plankton net, (b) fish net, (c) Niskin water sampler, (d) bathyphotometer, (e) sediment traps, (g) photomultiplier module, (f) autonomous underwater vehicles, (h) astrophysics optical modules ANTARES, (i-j) remotely operated vehicles.

Table 1: List of luminous bacterial species found in light organ symbiosis. In blue, the light organ position on the host body, according to the schema of fish from Nealson and Hastings, 1979. * firstly identified as *Vibrio logei* by Fidopiastis et al., 1998.









| Species | Host Collection | Hosts | Light Organ Location |
|---|---|---|---|
| *Aliivibrio fischeri* (*Vibrio fischeri*) | ***Euprymna* spp.** Western Pacific (Fidopiastis et al., 1998) <br><br> ***Sepiola* spp.** Mediterranean Sea, European Atlantic coast, Japan, Philippines (Fidopiastis et al., 1998) <br><br> ***Moconcentris japonica*** Japan (Dunlap et al., 2007) | **SEPIOLIDAE**   ***Euprymna* spp.**    *E. morsei*    *E. berryi*    *E. scolopes*    *E. tasmanica* <br> ***Sepiola* spp.**    *S. affinis*    *S. atlantica*    *S. intermedia*    *S. ligulata*    *S. robusta* |  |
|  | ***Cleidopus gloriamaris*** East coast of Australia (Fitzgerald, 1977) <br><br> ***Caelorinchus* spp.** Taiwan (*C. formosanus*) Japan (*C. multispinulosus*) (Dunlap et al., 2007) | **MONOCENTRIDAE**   ***Monocentris* spp.**    *M. japonica*   ***Cleidopus* spp.**    *C. gloriamaris* <br> **MACROURIDAE**   ***Caelorinchus* spp.**    *C. formosanus*    *C. multispinulosus* |  |
| *Aliivibrio thorii* | ***Sepiola affinis*** Mediterranean Sea (Fidopiastis et al., 1998 ; Ast et al., 2007) | **SEPIOLIDAE**   ***Sepiola* spp.**    *S. affinis* |  |
| *Aliivibrio wodanis*[*] | ***Sepiola* spp.** Mediterranean Sea (Fidopiastis et al., 1998 ; Ast et al., 2007) | **SEPIOLIDAE**   ***Sepiola* spp.**    *S. affinis*    *S. robusta* |  |
| *Photobacterium kishitanii* | ***Opisthoproctus* spp.** Atlantic Ocean (*O. grimaldii*) Atlantic Ocean and Indian Ocean (*O. soleatus*) (Haygood et al., 1992; Dunlap et al., 2007) <br><br> ***Chlorophthalmus* spp.** Japan (Dunlap et al., 2007) <br><br> ***Caelorinchus* spp.** Taiwan (*C. kishinouyei*) Japan (Other species) (Dunlap et al., 2007) <br><br> ***Malacocephalus laevis*** Indian Ocean (Dunlap et al., 2007) <br><br> ***Ventrifossa* spp.** Japan (*V. garmani* and *V. longibardata*) Taiwan (*V. rhidodorsalis*) (Dunlap et al., 2007) <br><br> ***Physiculus japonicus*** Japan (Dunlap et al., 2007) <br><br> ***Aulotrachichthys prosthemius*** Japan (Ast and Dunlap, 2004) <br><br> ***Acropoma hanedai*** Taiwan (Kaeding et al., 2007; Dunlap et al., 2007) | **OPISTHOPROCTIDAE**   ***Opisthoproctus* spp.**    *O. grimaldii*    *O. soleatus* <br> **CHLOROPHTHALMIDAE**   ***Chlorophthalmus* spp.**    *C. acutifrons*    *C. albatrossis*    *C. nigromarginatus* <br> **MORIDAE**   ***Physiculus* spp.**    *P. japonicus* <br> **MACROURIDAE**   ***Caelorinchus* spp.**    *C. anatirostris*    *C. denticulatus*    *C. fasciatus*    *C. hubbsi*    *C. japonicus*    *C. kamoharai*    *C. kishinouyei* <br> ***Malacocephalus* spp.**    *M. laevis*[*] <br> ***Ventrifossa* spp.**    *V. garmani*    *V. longibarbata*    *V. rhipidorsalis* <br> **TRACHICHTHYIDAE**   ***Aulotrachichthys* spp.**    *A. prosthemius* <br> **ACROPOMATIDAE**   ***Acropoma* spp.**    *A. hanedai* |  |





| Species | Host Collection | Hosts | Light Organ Location |
|---|---|---|---|
| *Photobacterium leiognathi* | *Acropoma japonicum*<br>Taiwan<br>(Kaeding et al., 2007) | **ACROPOMATIDAE**<br>*Acropoma* spp.<br>  *A. japonicum* | |
| | *Gazza* spp.<br>Philippines<br>(Dunlap et al., 2004, 2007) | **LEIOGNATHIDAE**<br>*Gazza* spp.<br>  *G. achlamys*<br>  *G. minuta* | |
| | *Leiognathus* spp.<br>Taiwan (*L. equulus*)<br>Okinawa (*L. fasciatus*)<br>Philippines (*L. jonesi, L. philippinus*)<br>Japan (*L. nuchalis*)<br>Gulf of Siam (*L. splendens*)<br>(Dunlap et al., 2004, 2007) | *Leiognathus* spp.<br>  *L. equulus*<br>  *L. fasciatus*<br>  *L. jonesi*<br>  *L. nuchalis*<br>  *L. philippinus*<br>  *L. splendens* | |
| | *Equulites* spp.<br>Japan (*E. elongatus, E. rivulatus*)<br>Philippines (*E. leucistus*)<br>(Dunlap et al., 2004, 2007) | *Equulites* spp.<br>  *E. elongatus*<br>  *E. leucistus*<br>  *E. rivulatus* | |
| | *Photopectoralis* spp.<br>Japan (*P. bindus*)<br>Philippines (*P. panayensis*)<br>(Kaeding et al., 2007) | *Photopectoralis* spp.<br>  *P. bindus*<br>  *P. panayensis* | |
| | *Photolateralis* spp.<br>Philippines (*P. stercorarius*)<br>(Dunlap et al., 2007) | *Photolateralis* spp.<br>  *P. stercorarius* | |
| | *Secutor* spp.<br>Philippines<br>(Dunlap et al., 2007) | *Secutor* spp.<br>  *S. insidiator*<br>  *S. megalolepis* | |
| | *Uroteuthis noctilus*<br>Sydney, Australia<br>(Guerrero-Ferreira et al., 2013) | **LOLIGINIDAE**<br>*Uroteuthis* spp.<br>  *U. noctiluca* | |
| | *Rondeletiola minor*<br>Mediterranean Sea, France<br>(Guerrero-Ferreira et al., 2013) | **SEPIOLIDAE**<br>*Rondeletiola* spp.<br>  *R. minor* | |
| | *Sepiolina nipponensis*<br>Japan<br>(Nishiguchi and Nair, 2003) | *Sepiolina* spp.<br>  *S. nipponensis* | |
| *Photobacterium mandapamensis* | *Acropoma japonicum*<br>Taiwan<br>(Kaeding et al., 2007) | **ACROPOMATIDAE**<br>*Acropoma* spp.<br>  *A. japonicum* | |
| | *Gadella jordani*<br>Taiwan<br>(Kaeding et al., 2007) | **MORIDAE**<br>*Gadella* spp.<br>  *G. jordani* | |
| | *Photopectoralis* spp.<br>Japan (*P. bindus*)<br>Philippines (*P. panayensis*)<br>(Kaeding et al., 2007) | **LEIOGNATHIDAE**<br>*Photopectoralis* spp.<br>  *P. bindus*<br>  *P. panayensis* | |
| | *Siphamia versicolor*<br>Japan<br>(Kaeding et al., 2007) | **APOGONIDAE**<br>*Siphamia* spp.<br>  *S. versicolor* | |
| *Vibrio harveyi* | *Uroteuthis chinensis*<br>Thailand<br>(Guerrero-Ferreira et al., 2013) | **LOLIGINIDAE**<br>*Uroteuthis* spp.<br>  *U. chinensis* | |
| | *Euprymna hyllbergi*<br>Thailand<br>(Guerrero-Ferreira et al., 2013) | **SEPIOLIDAE**<br>*Euprymna* spp.<br>  *E. hyllebergi* | |

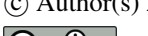



| Species | Host Collection | Hosts | Light Organ Location |
|---|---|---|---|
| *Candidatus* Enterovibrio escacola | *Ceratias* spp.<br>NE Atlantic (*C. sp*)<br>Gulf of Mexico (*C. uranoscopus*)<br><br>*Lynophryne maderensis*<br>NE Atlantic<br><br>*Melanocetus johnsoni*<br>Gulf of Mexico and NE Atlantic<br><br>*Melanocestus murrayi*<br>Gulf of Mexico<br><br>*Chaenophryne* spp.<br>NE Atlantic<br><br>*Oneirodes* sp.<br>Gulf of Mexico<br><br>(Baker et al., 2019) | **CERATIIDAE**<br>  *Ceratias* spp.<br>    *C. uranoscopus*<br>    *C. sp*<br><br>**LINOPHRYNIDAE**<br>  *Linophryne* spp.<br>    *L. maderensis*<br><br>**MELANOCETIDAE**<br>  *Melanocetus* spp.<br>    *M. johnsoni*<br>    *M. murrayi*<br><br>**ONEIRODIDAE**<br>  *Chaenophryne* spp.<br>    *C. longiceps*<br>    *C. sp*<br>  *Oneirodes* spp.<br>    *O. sp* | (illustration) |
| *Candidatus* Enterovibrio luxaltus | *Cryptopsaras couesii*<br>Gulf of Mexico and NE Atlantic<br>(Baker et al., 2019) | **CERATIIDAE**<br>  *Cryptopsaras* spp.<br>    *C. couesii* | (illustration) |
| *Candidatus* Photodesmus blepharus | *Photoblepharon* spp.<br>Pacific Ocean (*P. palpebratus*)<br>Western Indian Ocean (*P. steinitzi*)<br>(Hendry and Dunlap, 2014) | **ANOMALOPIDAE**<br>  *Photoblepharon* spp.<br>    *P. palpebratus*<br>    *P. steinitzi* | (illustration) |
| *Candidatus* Photodesmus katoptron | *Anomalops* spp.<br>Philippines<br>(Hendry and Dunlap, 2011) | **ANOMALOPIDAE**<br>  *Anomalops* spp.<br>    *A. katoptron* | (illustration) |