# Peer review of "Reviews and syntheses: Bacterial bioluminescence – ecology and impact in the biological carbon pump"

_Biogeosciences, 2020_

## Referee Comment (RC1) · Anonymous Referee #1 · 14 Apr 2020

This manuscript presents a very thorough review of the ecology of luminous marine bacteria in a variety of habitats (symbiosis, free-living, enteric). The paper is quite ambitious in scope and the authors have synthesized a lot of literature. Furthermore, the authors present a hypothesis that interactions of luminous bacteria with animal hosts may have important consequences for marine ecosystem level processes such as the biological carbon pump. It's hard to find this argument convincing because there is little known about luminous bacteria in many parts of this particular cycle, but I find the ideas presented very interesting and the authors have done an impressive job supporting their ideas with published literature and suggesting ideas for future research. The manuscript is generally well written, the figures are lovely, and I enjoyed reading it. The ambitious nature of the review makes it very long and sometimes hard to follow.

[Figure]

Because the authors are trying to review everything, some points seem out of place. I have made suggestions below for potential ways to shorten, focus and structure the manuscript to make it a bit easier to follow. My additional major comment is that in trying to provide a very broad review of all bioluminescent symbioses, the authors have sometimes given the impression that patterns found in one well studied symbiosis (E. scolopes - A. fischeri) are true of all bioluminescent symbioses. At points the authors fail to clarify when less (or nothing) is known from other systems, but we should not make the assumption that what is true for squid is generally true for other species. At other points, some data is available for fish systems, but it is sometimes missing from the manuscript or presented unevenly compared to squid work, as an add on or exception. I've made suggestions below for some additional references to consider and places to change wording to more evenly cover various luminous symbiotic systems.

General comments:

Lines 30-31 - I'd like references for the statements "luminous bacteria are the most abundant and are widely distributed" and "Most of the 30 currently known bacterial luminous species." What metrics are you using to say that luminous bacteria are more abundant and widespread than other luminous organisms? Abundant by biomass or prevalence? This seems like an unnecessary comparison in either case, since the ecology of bacteria is so different than luminous eukaryotes and they are likely using light in different ways. Maybe change this statement to something more general about the diversity and prevalence of luminous bacteria? Also, with the statement of a specific number of luminous species, citations need to be provided for these, such as a review with additional newer papers. Does this statement include terrestrial bacteria? I counted up the marine species I was aware of and didn't get 30, so the references would be useful for researchers in the field.

Lines 34 - 35- benefices change to benefits? I think these sentences could be clarified. What are the benefits of symbiosis to luminous bacteria? What are hypothesized benefits of luminescence to free-living bacteria? Why do you think that the carbon

pump may be important to this? Maybe a more general statement about the effects of bacterial luminescence on ecosystem level processes, such as the carbon pump, are understudied? The abstract does a good job walking the reader through how these very different ideas (luminescence, symbiosis and carbon cycling) are connected, but this is currently less well explained in the introduction and the transition to explain the carbon pump is awkward. In order to understand your arguments the reader has to understand that luminous bacteria are being released into the ocean from symbiosis of growth in guts and not all readers will be familiar with these facts. I think some of the ideas need to be stated earlier in the intro, which some examples and citations.

Lines 37-41 - The end point of the biological carbon pump is sequestration of carbon in ocean sediment, correct? I think this needs to be clearly stated here to explain that any marine snow that doesn't sink is being taken out of the pump.

Lines 94 - 98 - This should be restated that fish and squid with ventral light organs likely use them for counter illumination. As far as I'm aware, this has only been demonstrated for bobtailed squid, but is hypothesized in other cases where the light organ illuminates the animal's ventral surface. This is distinct from other fish which have light organs located externally and near the face. Also, some references on anomalopid behavior which might be useful: Morin et al., 1975, A light for all reasons, versatility in the behavioral repertoire of the flashlight fish; Hellinger et al., 2017, The Flashlight Fish Anomalops katoptron Uses Bioluminescent Light to Detect Prey in the Dark.

Lines 103 - 109 - Move the statement about the best studied symbiosis being that between Aliivibrio fischeri and E. scolopes to proceed these references and state that we don't understand how symbioses are established in most other systems. All of the references on light organ morphogenesis are on bobtailed squid and we don't know if similar mechanisms exist in most fish, so it's misleading to say that these things are common. For some references on light organ development and potential specificity factors in fishes see: Dunlap et al, 2013, Inception of bioluminescent symbiosis in early developmental stages of the deep-sea fish, Coelorinchus kishinouyei (Gadiformes: Macrouridae); Dunlap et al., 2012, Symbiosis initiation in the bacterially luminous sea urchin cardinal fish Siphamia versicolor; Gould and Dunlap, 2019, Shedding Light on Specificity: Population Genomic Structure of a Symbiosis Between a Coral Reef Fish and Luminous Bacterium

Lines 122 - 130 - I think this section is worded in a way that may be misleading. Light organs are generally monospecific, but not necessarily monoclonal, which is what the comparison to pure culture suggests to me. It's pretty well established that E. scolopes can be colonized by multiple strains (I think this is different from the wording here, "have been reported for some", which implies that multi strain colonization might happen but isn't common) (See several Bongrand and Ruby references such as https://www.nature.com/articles/s41396-018-0305-8) and similar levels of diversity seem to exist for some fish (I think some Dunlap references show multiple strains from a light organ, the Gould reference mentioned above discusses diversity with Siphamia light organs). Some fish do seem to have monoclonal light organs (Anomalopids and Ceratioids, Hendry et al, 2016, Genome Evolution in the Obligate but Environmentally Active Luminous Symbionts of Flashlight Fish, GBE; Baker et al., 2019). The wording for the Keading reference is also misleading, because not all of the fish studied in there had both symbionts. Please rephrase this section to more clearly state what is known for which species.

Line 169 - "Variation of light emission is closely linked to the concentration of one component involved in the bacterial light reaction, which could be host controlled" I'm not sure what the component being referred to here is, please explain and provide a reference.

Lines 166-173 - After this discussion of quorum sensing control in A. fischeri, it would be good to add mentions that it is not known if other species have similar control mechanisms, or the extent to which other host species control their symbionts. This review is very ambitious and I think trying to be very thorough, but as a consequence any missing information stands out. Be careful throughout to clarify what is known from only the

squid-vibrio system and what might be a common feature across host species. For instance, anomalopid symbionts have lost quorum sensing genes so that luminescence appears to be constitutively expressed in the bacteria (Hendry et al 2014; Hendry et al., 2016, GBE), and anglerfish symbionts don't have quorum sensing genes (Hendry et al 2016, mBio).

Lines 178 - 183. Again, these sentences are written as though they describe growth in light organs broadly but really describe what we know about the squid symbiosis. Please clarify that this may not be the situation for other host species. For instance, the Haygood 1984 reference that you use in the paragraph shows that monocentrids and anomalopids regularly release bacteria, rather than expelling them once a day. There are a number of differences between these systems which might account for this. These light organs are external, so bacteria can be pushed directly out of the tubules into sea water. Anomalopids are also strictly nocturnal and photophobic, they don't experience the same diurnal cycle that Euprymna does because they avoid light, so the same strategy of emptying the light organ and regrowing the bacteria may not be appropriate. Although much of the information in this review necessarily comes from the Euprymna system, in order to make it inclusive of bioluminescent symbiosis broadly, please be sure to compare and contrast what is known in other systems, or at the very least clarify when data from diverse systems is missing. It may be the case that in most symbiotic systems (fish), symbionts are released regularly and that the squid system is actually the exception, where there is one release per day. Currently, you mention these differences in a short paragraph (lines 193-195), but this feels like an add on, not an integrated part of the review that really tells us what is known and what is unknown.

Lines 213-215 - This discussion of P. leiognathi vs. V. harveyi seems unnecessary for the story, the point is just that fish guts have bioluminescent bacteria. The review is already fairly long and dense, I think this bit could be cut. Additionally, identification at the time would be difficult without the molecular sequencing abilities that we have now

to determine bacterial species.

Lines 228 - 265 - Similarly, I would suggest cutting some of these points about luminous bacteria in fish guts if they are not needed to support your points. The point you are trying to make, that fish gut content contribute to introducing luminous bacteria into sea water, is relatively straight forward and I'm not sure that the additional detail is needed. This whole section feels long to me. Note also that they Freed et al, 2019 reference includes discussion of ceratioid microbiome, including gut samples, which might be relevant.

Section 3.2 - It's not clear to me what role this section plays in the manuscript. As I said above, the review is aiming to be impressively thorough, but is becoming a little diffuse at points and a bit long. It's not really possible to include everything in a manuscript while keeping it manageable for the reader, so maybe consider if this is important information that the reader needs to know? This section is coming 8 pages into the text, out of an 18 page document, and we haven't yet gotten to the meat of the argument on the carbon pump, which is supposed to be a main focus of the paper. I think keeping the review a bit more focused with help the reader and highlight the new and interesting contributions of this paper.

The references that are just in Table 1 don't seem to be in the reference list. For example, Baker et al., 2019; Hendry and Dunlap, 2014; Hendry and Dunlap, 2011;

Specific comments:

Line 57 - Fig 1 is really nice, but I think its too complicated to ask the reader to look at this early in the manuscript, it seems like it would be referenced for the first time after some of these ideas have been introduced, in section 4.4.

Line 91 - internal, ventrally located

Lines 92-93 - this sentence is hard to follow, please rephrase

Lines 119 - 121 - This sentence is poorly worded, please revise.

Lines 121 - clarify that you mean bacterial species

Lines 131 - 134 - Some wording changes for clarity - "appears consistent at the host species level" to clarify host species tend to have one symbiont species, but symbiont species can colonize multiple host species. I don't understand this statement: "These symbiont strains present no clear phylogenetic divergence between themselves." Do you mean that host and symbiont phylogenies are not congruent?

Line 145 - Hendry et al., 2016 (GBE) is the genome description for the second anomalopid symbiont.

Line 149 - obligately dependent, not obligatory

Line 153 - I'm not sure what the sentence "The light organ is a separate and highly evolved entity" is referring to.

Line 154 - I don't think you want "communicate" here, maybe connect to? Or provide access to? Communicate implies that the bacteria are getting information from the light organ surface through the tubules, and I'm not sure that is known.

Line 156 - What is mechanical stimulation?

Line 339 - reword "the copiotrophic type"

Line 342 - "all . . . Vibrio and Photobacterium" I think this statement could be changed to something like "all luminous Vibrionaceae, except reduced genome symbionts, possess.." and still be accurate? I'm not aware of any Vibrionaceae species shown to just have 1 chromosome and the only examples of low rRNA operon copies that I know of are anomalopid and ceratioid symbionts. Not sure about Salinivibrio off the top of my head though. . .

Line 351 - Henceforth means "from now on," I think you want "therefore" or "hence"

Section 5.2.2 - This header is long and hard to follow, change to: quantification and diversity of luminous bacteria and their variability between ecosystems (free-living in

the water column, on sinking particles and fecal pellets, or in sediments)

Section 5.2.4 - What is lock in this context?

---

## Author Comment (AC1) · 28 Apr 2020

This manuscript presents a very thorough review of the ecology of luminous marine bacteria in a variety of habitats (symbiosis, free-living, enteric). The paper is quite ambitious in scope and the authors have synthesized a lot of literature. Furthermore, the authors present a hypothesis that interactions of luminous bacteria with animal hosts may have important consequences for marine ecosystem level processes such as the biological carbon pump. It's hard to find this argument convincing because there is little known about luminous bacteria in many parts of this particular cycle, but I find the ideas presented very interesting and the authors have done an impressive job supporting their ideas with published literature and suggesting ideas for future research.

The manuscript is generally well written, the figures are lovely, and I enjoyed reading it. The ambitious nature of the review makes it very long and sometimes hard to follow.

Because the authors are trying to review everything, some points seem out of place. I have made suggestions below for potential ways to shorten, focus and structure the manuscript to make it a bit easier to follow. My additional major comment is that in trying to provide a very broad review of all bioluminescent symbioses, the authors have sometimes given the impression that patterns found in one well studied symbiosis (E. scolopes - A. fischeri) are true of all bioluminescent symbioses. At points the authors fail to clarify when less (or nothing) is known from other systems, but we should not make the assumption that what is true for squid is generally true for other species. At other points, some data is available for fish systems, but it is sometimes missing from the manuscript or presented unevenly compared to squid work, as an add on or exception.

I've made suggestions below for some additional references to consider and places to change wording to more evenly cover various luminous symbiotic systems.

Answer: We thank Referee #1 for perceptive and helpful comments and will work to improve our manuscript. Indeed, in addition to a comprehensive review of the ecology of marine bioluminescent bacteria, our main goal is to present the link between bioluminescence and its potential impact on the biological carbon pump. Below, in blue, we highlight the modifications to our manuscript and discuss our responses to its suggestions. Along the text some parts that were not essential to our approach will be removed in order to lighten the text.

**General comments:**

Lines 30-31 - I'd like references for the statements "luminous bacteria are the most abundant and are widely distributed" and "Most of the 30 currently known bacterial luminous species." What metrics are you using to say that luminous bacteria are more abundant and widespread than other luminous organisms? Abundant by biomass or prevalence? This seems like an unnecessary comparison in either case, since the ecology of bacteria is so different than luminous eukaryotes and they are likely using light in different ways. Maybe change this statement to something more general about the diversity and prevalence of luminous bacteria? Also, with the statement of a specific number of luminous species, citations need to be provided for these, such as a review with additional newer papers. Does this statement include terrestrial bacteria?

I counted up the marine species I was aware of and didn't get 30, so the references would be useful for researchers in the field.

Answer: We agree with the reviewer that the notion of "abundance" is inappropriate in this context, and we will change the sentence for a more general statement talking about the prevalence of luminous bacteria: "Amongst marine light-emitting organisms, luminous bacteria

are the most widely distributed in oceans". Regarding the number of 30 bacterial luminous species, we referred to a synthesis on bacterial bioluminescence written by Dunlap (2014)*, in which the author talks about "Thirty or more species" and provides a table of species names. We will rephrase as follows: "Most of the currently known bacterial luminous species (about thirty) are heterotrophic, copiotrophic and facultatively anaerobic (Dunlap, 2014)."

*Dunlap, P. (2014). Biochemistry and genetics of bacterial bioluminescence. In Bioluminescence: Fundamentals and Applications in Biotechnology-Volume 1 (pp. 37-64). Springer, Berlin, Heidelberg.

Lines 34 - 35- benefices change to benefits? I think these sentences could be clarified.
What are the benefits of symbiosis to luminous bacteria? What are hypothesized benefits of luminescence to free-living bacteria? Why do you think that the carbon pump may be important to this? Maybe a more general statement about the effects of bacterial luminescence on ecosystem level processes, such as the carbon pump, are understudied? The abstract does a good job walking the reader through how these very different ideas (luminescence, symbiosis and carbon cycling) are connected, but this is currently less well explained in the introduction and the transition to explain the carbon pump is awkward. In order to understand your arguments the reader has to understand that luminous bacteria are being released into the ocean from symbiosis of growth in guts and not all readers will be familiar with these facts. I think some of the ideas need to be stated earlier in the intro, which some examples and citations.
Answer: As suggested, we will revise this part of the introduction section to elaborate a better connection between the different ideas that will be developed in the following sections.
We will rephrase as follows:
"[...] Bioluminescent species are found in most phyla from fish to bacteria (Haddock et al., 2010; Widder, 2010). Amongst marine light-emitting organisms, luminous bacteria are widely distributed in oceans. Most of the currently known bacterial luminous species (about thirty) are heterotrophic, copiotrophic and facultatively anaerobic (Dunlap, 2014). Endowed with important motility and chemotactic abilities, luminous bacteria are able to colonize a large variety of habitats (as symbionts with macro-organisms, free-living in seawater or attached to particles) (e.g. (Dunlap and Kita-tsukamoto, 2006) and references therein). In their symbiotic forms, bioluminescent bacteria are mostly known to colonize light organs and guts, in which they find better growing conditions than in the open ocean. These symbioses lead to a continuous release of luminous bacteria from light organs and digestive tracts, directly into the seawater or through fecal pellets (Ramesh et al., 1990). Bacterial bioluminescence in its free or attached forms is much less studied but is worth reconsidering, in its prevalence as well as its ecological implications. Indeed, some studies pointed out the well-adapted vision of fish or crustacean to the detection of point-source bioluminescence (Busserolles and Marshall, 2017; Frank et al., 2012; Warrant and Locket, 2004). The compiled data, from all forms of marine bacterial bioluminescence, presented and discussed in this review bring out the uninvestigated pathway of the bioluminescence contribution into the biological carbon pump, through the visual attraction of consumers for luminous particles.."

Lines 37-41 - The end point of the biological carbon pump is sequestration of carbon in ocean sediment, correct? I think this needs to be clearly stated here to explain that any marine snow that doesn't sink is being taken out of the pump.
Answer: We agree with the reviewer's comment and the sentence will be modified as follows:
"The biological carbon pump is defined as the process through which photosynthetic organisms convert $CO_2$ to organic carbon, as well as the export and fate of the organic carbon sinking from the surface layer to the dark ocean and its sediments by different pathways."

Lines 94 - 98 - This should be restated that fish and squid with ventral light organs likely use them for counter illumination. As far as I'm aware, this has only been demonstrated for bobtailed squid, but is hypothesized in other cases where the light organ illuminates the animal's ventral surface. This is distinct from other fish which have light organs located externally and near the face. Also, some references on anomalopid behavior which might be useful: Morin et al., 1975, A light for all reasons, versatility in the behavioral repertoire of the flashlight fish; Hellinger et al., 2017, The Flashlight Fish Anomalops katoptron Uses Bioluminescent Light to Detect Prey in the Dark.

Answer: We understand the comment and will reword this paragraph for clarity. It is true that there are studies demonstrating the counterillumination strategy for many species other than the bobtail squid (remaining the most commonly studied). These studies include non-bacterial bioluminescence.

Some references hereafter:

- Paitio, et al (2020). Reflector of the body photophore in lanternfish is mechanistically tuned to project the biochemical emission in photocytes for counterillumination.
- Claes et al (2010). Phantom hunter of the fjords: camouflage by counterillumination in a shark (Etmopterus spinax).
- Johnsen et al (2004). Propagation and perception of bioluminescence: factors affecting counterillumination as a cryptic strategy.
- Warner et al (1979). Cryptic bioluminescence in a midwater shrimp.

If we consider only luminous organisms in symbiosis with bacteria, the counterillumination strategy has been demonstrated for the bobtail squid and leiognathids fish, and hypothesized for others.

- Jones, B. W. and Nishiguchi, M. K.: Counterillumination in the Hawaiian bobtail squid, Euprymna scolopes Berry (Mollusca: Cephalopoda), Mar. Biol., 144(6), 1151–1155, https://doi.org/10.1007/s00227-003-1285-3, 2004.
- McFall-Ngai, M. J. and Morin, J. G.: Camouflage by disruptive illumination in Leiognathids, a family of shallow-water, bioluminescent fishes, J. Exp. Biol., 156(1), 119–137, 1991
- Dunlap, P. V., Kojima, Y., Nakamura, S. and Nakamura, M.: Inception of formation and early morphogenesis of the bacterial light organ of the sea urchin cardinalfish, Siphamia versicolor, Mar. Biol., 156(10), 2011–2020, https://doi.org/10.1007/s00227-009-1232-z, 2009.
- McAllister, D. E.: The significance of ventral bioluminescence in fishes, J. Fish. Res. Board Canada, 24(3), 537–554, https://doi.org/10.1139/f67-047, 1967.

This has been clarified in the text. Moreover, additional references have been added for other possible uses of bacterial bioluminescence in symbioses.

We will rephrase as follows: "Symbiotic luminescence seems more common in benthic or coastal environments for fish and squid as well (Haygood, 1993; Lindgren et al., 2012; Paitio et al., 2016). Shallow-water fishes with luminous bacterial symbionts include flashlight fishes (Anomalopidae), ponyfishes (Leiognathidae) and pinecone fishes (Monocentridae) (Davis et al., 2016; Morin, 1983). For deep-sea fishes, anglerfishes (Ceratiodei) and cods (Moridae) are among the common examples of luminous-bacteria hosts.

Bacterial and intrinsic light organs are predominantly internal, ventrally located (Paitio et al., 2016). Many luminous organisms with ventral light organs likely use the emitted light to conceal themselves by counterillumination. This defensive strategy allows luminous species to match with the intensity, spectrum, and angular distribution of the downwelling light, thus obliterating their silhouette and therefore avoiding dusk-active piscivorous predators (Claes et al., 2010; Johnsen et al., 2004; Warner et al., 1979). Amongst bacterial light symbioses, counterillumination has been demonstrated for the bobtail squid Euprymna scolopes (Jones and Nishiguchi, 2004), some leiognathids fish (McFall-Ngai and Morin, 1991), and hypothesized for other bioluminescent fishes (Dunlap et al., 2009; McAllister, 1967). Less common but more

striking, some organisms found in the families Monocentridae, Anomalopidae and numerous deep-sea anglerfishes belonging to the suborder Ceratoidei, exhibit externally-located light organs colonized by bacteria (Haygood, 1993). The external light organs of flashlight fish have been demonstrated to be used to illuminate nearby environment and detect prey (Hellinger et al., 2017), or schooling behavior (Gruber et al., 2019), while the lure of female anglerfish is generally believed to be used for mate-finding purposes and prey attraction (Herring, 2007)."

Lines 103 - 109 - Move the statement about the best studied symbiosis being that between Aliivibrio fischeri and E. scolopes to proceed these references and state that we don't understand how symbioses are established in most other systems. All of the references on light organ morphogenesis are on bobtailed squid and we don't know if similar mechanisms exist in most fish, so it's misleading to say that these things are common. For some references on light organ development and potential specificity factors in fishes see: Dunlap et al, 2013, Inception of bioluminescent symbiosis in early developmental stages of the deep-sea fish, Coelorinchus kishinouyei (Gadi- formes: Macrouridae); Dunlap et al., 2012, Symbiosis initiation in the bacterially luminous sea urchin cardinal fish Siphamia versicolor; Gould and Dunlap, 2019, Shedding Light on Specificity: Population Genomic Structure of a Symbiosis Between a Coral Reef Fish and Luminous Bacterium

Answer: As suggested, the statement about the squid-*Vibrio* symbiosis constituting the major source of information for luminous symbiosis has been moved at the beginning of paragraph 2.2. The paragraph will be lightened to improve clarity. A sentence will be added to answer the reviewer's comment as follows:

"While the bobtail-squid model provides a window to understand the establishment of such symbioses, this system cannot be systematically transferred to other bacterial luminous symbioses. Although less well known, the other associations are no less important and many questions remain unresolved since they might be harder to study."

Throughout the text, we have been cautious to specify when our point was to specifically discuss the bobtail squid symbiosis. As examples:

"One of the best-documented symbioses is the association of *Aliivibrio fischeri* with the bobtail squid *Euprymna scolopes* [...]."

"Knowledge of the mechanisms involved in the selection and the establishment of bacterial symbionts in the squid-*Vibrio* symbiosis have considerably improved over the last few decades."

Lines 122 - 130 - I think this section is worded in a way that may be misleading. Light organs are generally monospecific, but not necessarily monoclonal, which is what the comparison to pure culture suggests to me. It's pretty well established that E. scolopes can be colonized by multiple strains (I think this is different from the wording here, "have been reported for some", which implies that multi strain colonization might happen but isn't common) (See several Bongrand and Ruby references such as https://www.nature.com/articles/s41396-018-0305-8) and similar levels of diversity seem to exist for some fish (I think some Dunlap references show multiple strains from a light organ, the Gould reference mentioned above discusses diversity with Siphamia light organs). Some fish do seem to have monoclonal light organs (Anomalopids and Ceratioids, Hendry et al, 2016, Genome Evolution in the Obligate but Environmentally Active Luminous Symbionts of Flashlight Fish, GBE; Baker et al., 2019). The wording for the Keading reference is also misleading, because not all of the fish studied in there had both symbionts. Please rephrase this section to more clearly state what is known for which species.

Answer: The paragraph will be removed since it was not essential in our approach. It allows lightening the text.

Line 169 - "Variation of light emission is closely linked to the concentration of one component involved in the bacterial light reaction, which could be host controlled" I'm not sure what the component being referred to here is, please explain and provide a reference.

Answer: The component was referring to molecules like oxygen, iron or phosphate which concentrations can be regulated inside the light organ leading to extremely favorable conditions as explained at the end of the paragraph. However, we agree that this sentence was confusing and it will be removed from the new version.

Lines 166-173 - After this discussion of quorum sensing control in A. fischeri, it would be good to add mentions that it is not known if other species have similar control mechanisms, or the extent to which other host species control their symbionts. This review is very ambitious and I think trying to be very thorough, but as a consequence any missing information stands out. Be careful throughout to clarify what is known from only the squid-vibrio system and what might be a common feature across host species. For instance, anomalopid symbionts have lost quorum sensing genes so that luminescence appears to be constitutively expressed in the bacteria (Hendry et al 2014; Hendry et al., 2016, GBE), and anglerfish symbionts don't have quorum sensing genes (Hendry et al 2016, mBio).

Answer: A sentence will be added to specify that quorum-sensing is not a common feature, as follows: "Here again, while the control mechanisms of the squid-Vibrio symbiosis are well understood, these of the other symbioses remain enigmatic and there are indications that they may vary. For example, the absence of the quorum-sensing-gene detection in anglerfish and flashlight fish symbionts suggests a constitutive light emission by the bacteria (Hendry et al. 2016, 2018).".

Lines 178 - 183. Again, these sentences are written as though they describe growth in light organs broadly but really describe what we know about the squid symbiosis. Please clarify that this may not be the situation for other host species. For instance, the Haygood 1984 reference that you use in the paragraph shows that monocentrids and anomalopids regularly release bacteria, rather than expelling them once a day.
There are a number of differences between these systems which might account for this. These light organs are external, so bacteria can be pushed directly out of the tubules into sea water. Anomalopids are also strictly nocturnal and photophobic, they don't experience the same diurnal cycle that Euprymna does because they avoid light, so the same strategy of emptying the light organ and regrowing the bacteria may not be appropriate. Although much of the information in this review necessarily comes from the Euprymna system, in order to make it inclusive of bioluminescent symbiosis broadly, please be sure to compare and contrast what is known in other systems, or at
the very least clarify when data from diverse systems is missing. It may be the case that in most symbiotic systems (fish), symbionts are released regularly and that the squid system is actually the exception, where there is one release per day. Currently, you mention these differences in a short paragraph (lines 193-195), but this feels like an add on, not an integrated part of the review that really tells us what is known and what is unknown.

Answer: Thanks for this very important comment. We will modify the paragraph and reorganize it as follows:
"For all symbioses, luminous symbionts, within the light organ, reach a very high density which reduces the oxygen availability, essential for the light reaction. Such oxygen limitation leads to a decrease in the specific luminescence activity (Boettcher et al., 1996). Bacterial population inside the light organ is regulated by the host, by coupling the restriction of the growth rate and the expulsion of symbionts. Growth repression is thought to reduce the energetic cost of the symbiosis to the host (Haygood et al., 1984; Ruby and Asato, 1993; Tebo et al., 1979). Additionally, since luminous bacteria are densely packed inside tubules communicating with

the exterior of the light organ (Haygood, 1993), the cell number of symbionts is regulated by the regular expulsion of most of the bacterial population, followed by a period of regrowth of the remaining symbionts. Concerning the well-known squid-*Vibrio* symbiosis, its daily release is highly correlated with the diel pattern of the host behavior. Indeed, the bobtail squid expels 95 % of the luminous symbionts in the surrounding environment at dawn, the beginning of its inactive phase. The remaining 5 % of *A. fischeri* grow through the day and the highest concentration is reached at the end of afternoon, at the nocturnal active phase of the squid (Nyholm and McFall-Ngai, 2004; Ruby, 1996). Currently, with the exception of the squid-*Vibrio* symbiosis, accurate data on the symbiont release are still largely unknown. Indeed, the frequency of release may vary and occur more than once a day as it has been shown for some flashlight and pinecone fishes (Haygood, 1984)."

Lines 213-215 - This discussion of P. leiognathi vs. V. harveyi seems unnecessary for the story, the point is just that fish guts have bioluminescent bacteria. The review is already fairly long and dense, I think this bit could be cut. Additionally, identification at the time would be difficult without the molecular sequencing abilities that we have now to determine bacterial species.
Answer: Part of the paragraph will be removed since it was not essential in our approach. It allows lightening the text. The sentence will be as follows:
"Most hosts with internal light organ release luminous bacteria into the digestive tract (Haygood, 1993; Nealson and Hastings, 1979), and thus may largely contribute to their abundance in luminous fish intestines. However, many fishes without light organ also harbor luminescent bacteria in their gut (Makemson and Hermosa, 1999), which clearly demonstrates the existence of other sources for enteric luminous bacteria."

Lines 228 - 265 - Similarly, I would suggest cutting some of these points about luminous bacteria in fish guts if they are not needed to support your points. The point you are trying to make, that fish gut content contribute to introducing luminous bacteria into sea water, is relatively straight forward and I'm not sure that the additional detail is needed. This whole section feels long to me. Note also that they Freed et al, 2019 reference includes discussion of ceratioid microbiome, including gut samples, which might be relevant.
Answer: We agree that some of our explanations were straight forward for the microbiologist community. We will remove some sentences that were redundant. However, this article is dedicated to a pluridisciplinary audience and we decided to keep some parts that, we believe, will be helpful for non specialists. We will also add the reference of Freed et al (2019) relevant in this paragraph.

Section 3.2 - It's not clear to me what role this section plays in the manuscript. As I said above, the review is aiming to be impressively thorough, but is becoming a little diffuse at points and a bit long. It's not really possible to include everything in a manuscript while keeping it manageable for the reader, so maybe consider if this is important information that the reader needs to know? This section is coming 8 pages into the text, out of an 18 page document, and we haven't yet gotten to the meat of the argument on the carbon pump, which is supposed to be a main focus of the paper. I think keeping the review a bit more focused with help the reader and highlight the new and interesting contributions of this paper.
The references that are just in Table 1 don't seem to be in the reference list. For example, Baker et al., 2019; Hendry and Dunlap, 2014; Hendry and Dunlap, 2011
Answer: This section will be deleted to reduce the length of the manuscript. The missing references will be added.

**Specific comments:**

Line 57 - Fig 1 is really nice, but I think it's too complicated to ask the reader to look at this early in the manuscript, it seems like it would be referenced for the first time after some of these ideas have been introduced, in section 4.4.
Answer: We discussed while writing the interest of putting the figure 1 at the end of the introduction. We thought that it would be easier for the reader to be able to use it as a guideline throughout the review and modified our text to say so.
We will add the following sentence: "Figure 1 represents, throughout the text, the guideline of the bioluminescence shunt hypothesis of the biological carbon pump."

Line 91 - internal, ventrally located
Answer: We will rephrase as follow: "Bacterial and intrinsic light organs are predominantly internal, ventrally located (Paitio et al., 2016)"

Lines 92-93 - this sentence is hard to follow, please rephrase
Answer: The sentence has been removed since it was not essential in our approach. It allows lightening the text.

Lines 119 - 121 - This sentence is poorly worded, please revise.
Answer: The sentence will be removed.

Lines 121 - clarify that you mean bacterial species
Answer: "Bacterial" will be added.

Lines 131 - 134 - Some wording changes for clarity - "appears consistent at the host species level" to clarify host species tend to have one symbiont species, but symbiont species can colonize multiple host species. I don't understand this statement: "These symbiont strains present no clear phylogenetic divergence between themselves." Do you mean that host and symbiont phylogenies are not congruent?
Answer: The paragraph will be removed since it was not essential in our approach. It allows lightening the text.

Line 145 - Hendry et al., 2016 (GBE) is the genome description for the second anomalopid symbiont.
Answer: The reference Hendry et al., 2016 (GBE) will be added.

Line 149 - obligately dependent, not obligatory
Answer: It will be changed

Line 153 - I'm not sure what the sentence "The light organ is a separate and highly evolved entity" is referring to.
Answer: The sentence will be removed.

Line 154 - I don't think you want "communicate" here, maybe connect to? Or provide access to? Communicate implies that the bacteria are getting information from the light organ surface through the tubules, and I'm not sure that is known.

Answer: As suggested, "communicate to" will be replaced by "connect to".

Line 156 - What is mechanical stimulation?

Answer: This part will be removed in this section since unappropriated here. However, we think that it is important to specify the kinetic differences between luminous bacteria and other organisms, since we use this fundamental feature in section 5.2.1 of our manuscript. The mechanical stimulation notion is commonly used in the litterature. As an example, dinoflagellates emit light due to wave motion (a mechanical stimulation). So, we will add in the introduction section the following sentence:

"Luminescent bacteria can glow continuously under specific growth conditions (Nealson and Hastings, 1979), while, in contrast, eukaryotic bioluminescent organisms require mechanical stimulation to emit light (Haddock et al., 2010)."

Line 339 - reword "the copiotrophic type"

Answer: We reworded this sentence to 'the copiotrophic trait' which is more appropriated.

Line 342 - "all : : : Vibrio and Photobacterium" I think this statement could be changed to something like "all luminous Vibrionaceae, except reduced genome symbionts, possess.." and still be accurate? I'm not aware of any Vibrionaceae species shown to just have 1 chromosome and the only examples of low rRNA operon copies that I know of are anomalopid and ceratioid symbionts. Not sure about Salinivibrio off the top of my head though…

Answer: We agree with this suggestion and the sentence will be changed as suggested: "All luminous *Vibrionaceae*, except reduced genome symbionts, possess two chromosomes in their genome [...]"

Line 351 - Henceforth means "from now on," I think you want "therefore" or "hence"

Answer: As suggested, "Henceforth" will be replaced by "Hence".

Section 5.2.2 - This header is long and hard to follow, change to: quantification and diversity of luminous bacteria and their variability between ecosystems (free-living in the water column, on sinking particles and fecal pellets, or in sediments)

Answer: We have followed the reviewer's suggestion and we will modify the header as proposed. Moreover, in the next section (5.2.4), we will follow the same advice and will reduce both headers. The headers will be as follows:

5.2.2 Quantification and diversity of luminous bacteria and their variability between ecosystems (free-living in the water column, on sinking particles and fecal pellets, or in sediments)

5.2.4 Quantification of the particles consumption rate and fate of the organic matter between glowing and non-glowing particles

Section 5.2.4 - What is lock in this context?

Answer: We will modify the beginning of this subsection to clarify our goals.
This sentence will be removed:

"One main lock to evaluate the importance of bioluminescence in the biological carbon pump is to quantify the transfer rate of organic carbon between trophic levels."

And we will add a more detailed description as follow:
"One current challenge to evaluate the importance of bioluminescence in the biological carbon pump is that, in the literature, there is no quantification of organic carbon transfer rates due to glowing bacteria attached to particles to higher trophic levels. Comparisons between glowing particles and non-glowing ones and the fate of the organic matter (i.e. decomposition, and particles sinking rate and fluxes) in both cases are necessary."

---

## Referee Comment (RC2) · Anonymous Referee #2 · 24 May 2020

General comments:

This is a fascinating subject for a review and I read it with much interest. It is extremely thorough, and in some places even a bit too detailed and requires a step back for the non-expert (see specific comments below). It is well organized and generally well written, although requires a thorough editing for grammar (some examples below).

The one figure and Table are well done, but in a review of this detail and length a few more figures to help illustrate some of the concepts would be helpful. One example that comes to mind is a diagram showing the mechanisms of expulsion.

The discussion on impacts on the biological C pump need to be qualified more. Luminescent bacteria are not always a catalyst for sequestration. If bioluminescence leads

to disaggregation and "slowing down the sinking rate of particles and consequently increasing their degradation and the remineralization rate" and this happens in the mixed layer, that will decrease carbon export and sequestration.

Specific comments:

Paper uses 'bacteria' throughout. Are Archaea bioluminescent too? (This should be mentioned somewhere).

p. 2, L 34 beneficies (should this be benefits?) p. 3, L 68 spelling- evidence p. 3, L 77 pyrosomes are not fishes (they are pelagic tunicates) p. 3, L 87 Anglerfishes- would be more clear if you give the rule first then the exception (isn't it that nearly all the esca in Angler fishes are symbiotic luminous bacteria and not intrinsic light organs? p. 4, L 91 spelling- internal

Section 2.2; p. 4, L 101-118 This section gives examples, but does not actually explain how symbiont selection or colonization occurs. What is 'microbial recognition and molecular dialog' and how does it work? How colonization occurs is not described at all.

p. 6, L 174- spelling- reduces p. 6, L 176- The bacterial . . . p. 7, L 193- More detail needed here. How does the expulsion actually take place? How do the bacteria get from the tubules into the digestive tract (are all light organs directly connected to the digestive tract, and through what)? Or from tubules into the surrounding water, for that matter- do all tubules have an opening on the animal surface- seawater interface, or only some ? For example, I have always wondered in an Anger fish esca, how are the bacteria expulsed? A figure would be helpful to illustrate.

p. 7, L 193- "Most hosts with internal light organs. . ."

p.8, L237- "in an herbivorous fish compared to a carnivore." p.8, L240- prey

p. 9, L273- what is meant by 'A rare item'? Do you mean that one rare piece of information we do have is that luminescent bacteria are known to help in chitin digestion,

or that in rare cases luminescent bacteria are known to help in chitin digestion.

p.11 , L329-330 'prior eaten' is awkward

p.12 , L353 'and is always associated with luminous bacteria'

p.13, L387- replace the word 'unbelievable' p.13, L394- 'amphipods were attracted' p.13, L398- do you mean 'the attraction of luminous bacteria to zooplankton?

p.13, L404- replace 'excreted' with 'egested' p.13, L414- replace 'excreted' with 'egested'

p. 14, L424-429. As mentioned in general comments, need to be careful here- it is not always a catalyst for sequestration: if bioluminescence leads to disaggregation and slowing down the sinking rate of particles and consequently increasing their degradation and the remineralization rate , and this happens in the mixed layer, that will decrease carbon flux and sequestration.

p. 14, L438- relies p. 14, L448- replace 'pilled' with 'combines'

p. 15, L467- 'role of bioluminescence bacteria...' p. 15, L473- 'pursuit' of investigations p. 15, L475-476- be specific- vertical migration of what ? (diel vertical migration zooplankton and fish?)

p.16, L486-487; suggest make this more broad/ global statement than just European initiatives (mention of ARGO is good, and Bioargo should be mentioned too).

p. 17, L518- The 'pursuit' of investigations p. 17, L528- what about use of acrylamide gels in sediment traps, which preserve the integrity of the particle, and presumably the attached bacteria?

p. 17, section 5.2.3- I found this section unfocused (too much of 'catch all'), and it also does not discuss vertical migration, which is mentioned in the section heading.

p. 18, section 5.2.4 L554- the word 'lock' needs to be replaced

whole section- I thought bioluminescence in zooplankton was used mainly to startle or confuse a predator. Also, bacteria in fecal pellets should be mentioned in this section

Figure 1- not clear to me why the arrow in 4 denotes slow sinking (why are particles released from vertical migrators slower than those repackaged or from sloppy feeding?)

Table 1.- Caption should specify 'in fishes and squids' (as there are also luminescent bacteria in zooplankton, which are not shown here). "List of luminous bacterial species found in light organ symbiosis in fishes and squids"

---

## Author Comment (AC3) · 3 Jun 2020

**General comments:**
This is a fascinating subject for a review and I read it with much interest. It is extremely thorough, and in some places even a bit too detailed and requires a step back for the non-expert (see specific comments below). It is well organized and generally well written, although requires a thorough editing for grammar (some examples below).
The one figure and Table are well done, but in a review of this detail and length a few more figures to help illustrate some of the concepts would be helpful. One example that comes to mind is a diagram showing the mechanisms of expulsion.
The discussion on impacts on the biological C pump need to be qualified more. Luminescent bacteria are not always a catalyst for sequestration. If bioluminescence leads to disaggregation and "slowing down the sinking rate of particles and consequently increasing their degradation and the remineralization rate" and this happens in the mixed layer, that will decrease carbon export and sequestration.

Answer: We thank referee#2 for his favorable comments. We will have the manuscript proofread by a language specialist. We agree that a review such as ours would benefit from a little more illustration. However, we chose not to add the illustration suggested by the referee because the mechanisms of expulsion are little known and, as far as we know, differ from one organism to another. Indeed, there are numerous types of light organs, with a large diversity of both structure and location. Only a few of them have been described in detail. The most studied is that of the squid but, in accordance with the comments of referee #1, we have chosen to avoid systematically focusing our interest on this organism so as not to make its functioning a generality. However, in order to integrate additional information on the localisation of the ejections of bioluminescent bacteria, either directly into the surrounding seawater or indirectly through the gut, we will complete Table 1 (see at the end of this document). The Table caption will be changed as followed:

Table 1: List of luminous bacterial species found in light organ symbiosis in fishes and squids. The diagrammatic fish, from Nealson and Hastings (1979), was used to indicate, in blue, the approximate locations of the light organ of the different families of symbiotically-luminous fishes. E: indicates an external expulsion of the bioluminescent bacteria, directly into the seawater. I: indicates an internal expulsion of the bioluminescent bacteria, in the digestive tract. (E) or (I) indicate a putative localisation of the expulsion.

Moreover, we propose the addition of another illustration, that will explain in more detail the importance of bioluminescence in the accessibility of organic matter for marine organisms, in section 4.4.

[Figure]

Figure 2: Zoom on the carbon fluxes at the level of a gravitational sinking particle (inspired by Azam and Long 2001). The sinking POC is moving downward followed by the chemical plume (Kiørboe 2011). The plain white arrows represent the carbon flow. Panel (a) represents the classical view of a non-bioluminescent particle. The length of the plume is identified by the scale on the side (Kiørboe and Jackson 2001). Panel (b) represents the case of a glowing particle in the bioluminescence shunt hypothesis. Bioluminescent bacteria are represented aggregated onto the particle. Their light emission is shown as a bluish cloud around it. Blue dotted arrows represent the visual detection and the movement toward the particle of the consumer organisms. Increasing the visual detection allows a better detection by upper trophic levels, potentially leading to the fragmentation of sinking POC into suspended POC due to sloppy feeding. The consumption of the bioluminescent POC by fish can lead to the emission of bioluminescent fecal pellets (repackaging), which can also be produced with non-bioluminescent POC if the fish gut is already charged with bioluminescent bacteria.

**Specific comments:**

Paper uses 'bacteria' throughout. Are Archaea bioluminescent too? (This should be mentioned somewhere).
Answer: No archaea has been characterized as bioluminescent. The sentence "To our knowledge, no archaea has been characterized as bioluminescent" will be added in the introduction section.

p. 2, L 34 beneficies (should this be benefits?)
Answer: Done

p. 3, L 68 spelling- evidence
Answer: Done

p. 3, L 77 pyrosomes are not fishes (they are pelagic tunicates)
Answer: We will modify the paragraph to clarify. In this paragraph, we will discuss the symbioses with luminous bacteria in general and not only with fishes.

p. 3, L 87 Anglerfishes- would be more clear if you give the rule first then the exception (isn't it that nearly all the esca in Angler fishes are symbiotic luminous bacteria and not intrinsic light organs?
Answer: This part will be removed subsequently to the referee #1 comments in order to lighten the text.

p. 4, L 91 spelling- internal
Answer: Done

Section 2.2; p. 4, L 101-118 This section gives examples, but does not actually explain how symbiont selection or colonization occurs. What is 'microbial recognition and molecular dialog' and how does it work? How colonization occurs is not described at all.
Answer: This review is already very thorough as both referees commented. We would rather not add more information regarding subjects that are not directly related to the BCP since many authors have already extremely well reviewed information on symbiont selection or colonization and the more described are the squid's ones. These publications are indicated in the text. As suggested by referee #1, we don't want to talk systematically about the squid so that it doesn't become the general case. Moreover, the text will slightly be modified at some points in order to clarify what is known only for the squid symbiosis and what is valid for all symbioses. These changes are indicated in the reply to referee#1.

p. 6, L 174- spelling- reduces
Answer: Done

p. 6, L 176- The bacterial ...
Answer: Done

p. 7, L 193- More detail needed here. How does the expulsion actually take place? How do the bacteria get from the tubules into the digestive tract (are all light organs directly connected to the digestive tract, and through what)? Or from tubules into the surrounding water, for that matter- do all tubules have an opening on the animal surface- seawater interface, or only some ? For example, I have always wondered in an Anglerfish esca, how are the bacteria expulsed? A figure would be helpful to illustrate.
Answer: As mentioned above, there is an important diversity in the structure and location of the light organs, and actually, with the squid exception, many points of the other symbioses (symbiont selection, population regulation, frequency of the symbiont expulsion...) remain unclear. That's why it is not possible to have a simple description of the process. Since this is not the topic of our review and as explained in the 2.2 answer, we chose not to add a figure. However, this comment prompted us to add, in the Table 1, an information related to the expulsion pathway of the luminous bacteria (directly connected to the environment if the light organ has pores or ducts opening into the surrounding sea, or indirectly if the light organ has ducts connected to the gut). We think it is an interesting piece of information and thank the referee#2 for that.

p. 7, L 193- "Most hosts with internal light organs…"
Answer: Done.

p.8, L237- "in an herbivorous fish compared to a carnivore." p.8, L240- prey

Answer: Done

p. 9, L273- what is meant by 'A rare item'? Do you mean that one rare piece of information we do have is that luminescent bacteria are known to help in chitin digestion, or that in rare cases luminescent bacteria are known to help in chitin digestion.
Answer: The former suggestion is the right one. However, this section will be deleted to reduce the length of the manuscript according to referee #1 comments.

p.11 , L329-330 'prior eaten' is awkward
Answer: This part will be removed since this idea is already discussed all along the paragraph and the turn of phrase was not ideal.

p.12 , L353 'and is always associated with luminous bacteria'
Answer: Done

p.13, L387- replace the word 'unbelievable'
Answer: 'unbelievable' will be replaced by "huge".
The sentence will be as follows: "As indicated previously, the release of bioluminescent bacteria from light organs and fecal pellets could represent a huge quantity of bioluminescent bacteria in the water column."

p.13, L394 - 'amphipods were attracted'
Answer: Done

p.13, L398- do you mean 'the attraction of luminous bacteria to zooplankton'?
Answer: No, we mean the contrary. Since the sentence was confusing, the two last sentences of the paragraph will be modified as follows : "To our knowledge, the only one known is from Zarubin et al. (2012), who demonstrated that zooplankton is attracted to luminous particles and feeds on the luminous bacteria-rich organic matter. Because of the ingestion of the luminous bacteria, the zooplankton itself starts to glow. Then, they experimentally measured 8-times-higher ingestion rate of glowing (due to ingestion of bioluminescent bacteria) zooplankton by fishes, compared to non-luminous zooplankton."

p.13, L404- replace 'excreted' with 'egested'
Answer: Done

p.13, L414- replace 'excreted' with 'egested'
Answer: Done

p. 14, L424-429. As mentioned in general comments, need to be careful here- it is not always a catalyst for sequestration: if bioluminescence leads to disaggregation and slowing down the sinking rate of particles and consequently increasing their degradation and the remineralization rate, and this happens in the mixed layer, that will decrease carbon flux and sequestration.
Answer: We agree with the comment. Bioluminescence can impact the BCP in both ways and we clearly indicate these two hypotheses several times through the text. We realize that the term catalyst can be misinterpreted. We will modify the specific paragraph to clarify as follows :

"Considering this bioluminescence shunt hypothesis, all the processes described above show that bioluminescence affects  the biological gravitational carbon pump (Boyd et al., 2019), by either increasing the carbon sequestration into the deep ocean, or by slowing down the sinking rate of particles and consequently increasing their degradation and the remineralization rate. Bioluminescence and especially luminous bacteria may therefore influence the export and

sequestration of biogenic carbon in the deep oceans (either positively or negatively). A better quantification of these processes and impacts in the biological carbon pump are a requirement in future studies."

p. 14, L438- relies
Answer: Done

p. 14, L448- replace 'pulled' with 'combines'
Answer: Done

p. 15, L467- 'role of bioluminescence bacteria...'
Answer: In this subpart, we not only propose to investigate bioluminescent bacteria but more generally to quantify bioluminescence globally (as indicated for exemple in "1) the assessment of the global importance of bioluminescence in the oceans"). This justifies the use of a more general title.

p. 15, L473- 'pursuit' of investigations
Answer: Done

p. 15, L475-476- be specific- vertical migration of what ? (diel vertical migration zooplankton and fish?)
Answer:We will define the vertical migration more precisely as suggested.

p.16, L486-487; suggest make this more broad/ global statement than just European initiatives (mention of ARGO is good, and Bioargo should be mentioned too).
Answer: We agree with the comment. We will modify the text as follows :
'For temporal scales, in the last decades, the multiplication of long-term observatories such as Ocean Network Canada (ONC), the Ocean Observatories Initiative (OOI), the station ALOHA, the European Multidisciplinary Seafloor and water column Observatory (EMSO-ERIC), or the Biogeochemical Argo International Program have increased global-ocean observations at long time scales (more than 10 years) and high sampling frequency.'

p. 17, L518- The 'pursuit' of investigations
Answer: The section title will be changed according to the referee #1.

p. 17, L528- what about use of acrylamide gels in sediment traps, which preserve the integrity of the particle, and presumably the attached bacteria? Fecal pellets should be mentioned in this section
Answer: Acrylamide gel is efficient for the conservation of the pellets. It might be worth trying for cell conservation but will certainly alter the bioluminescence. For that reason, we decided not to add this methodology into the subsection.

p. 17, section 5.2.3- I found this section unfocused (too much of 'catch all'), and it also does not discuss vertical migration, which is mentioned in the section heading. Fecal pellets should be mentioned in this section

Answer: We will follow reviewer #2's comment and will remove this section. Two sentences will be moved into the next subsection 5.2.4, since we believe that this information, based on already existing literature, is of major importance for future investigations.

"As an example, Vibrio are important contributors to particulate organic carbon fluxes that have been observed at abyssal depths in the Pacific Ocean (Preston et al., 2019, Boeuf et al., 2019).

A better characterization at species or functional level should highlight the luminous potential related to the presence of such organisms, even at low abundance."

The description of the effects of vertical migration of zooplankton and fish on luminous bacteria dispersal will be added in part 4.4 (Figure 1, step 4), we will include the following details:

"Additionally, the consumption of organic material colonized by bioluminescent bacteria increases their dispersal rate provided by migrating zooplankton, and even more so by actively swimming fish, following the conveyor-belt hypothesis (Grossart et al., 2010) (Figure 1, step 4). After being ingested, bacteria (including luminous ones), attached to the particles consumed by zooplankton and fish, stay in their digestive tract. At night, these organisms migrate in the upper part of the water column and release feces in niches and at depth that, eventually, would not have been otherwise colonized by luminous bacteria. This dispersion, due to the expelling of luminous feces, is several orders of magnitude greater than that of water-borne free bacteria."

p. 18, section 5.2.4 L554- the word 'lock' needs to be replaced whole section- I thought bioluminescence in zooplankton was used mainly to startle or confuse a predator. Also, bacteria in fecal pellets should be mentioned in this section.

Answer: We will remove the word 'lock' and use "One current challenge". In this subsection we mainly described future actions to quantify the attraction rate of particles (including fecal pellets), glowing due to bioluminescent bacteria, by higher trophic levels. As the reviewer says, it is commonly admitted that bioluminescence from bacteria attracts, while flashes of light in most zooplankton deters. Here we describe the attraction of bacteria on zooplankton. We will add the sentence as follows to avoid misunderstanding and take into account the comment of the reviewer:

"Few studies related the preferential consumption of luminous bacteria by zooplankton (copepods in Nishida et al., 2002) or fish (Zarubin et al., 2012). It is well-known that marine snow is intensively colonized by bacteria (about $10^9$ bacteria per millilitre) (Azam & Long, 2001). Amongst them, luminous bacteria attract zooplankton by emitting light continuously (while flashes of light emitted by zooplankton deter, as mentioned earlier)."

Figure 1- not clear to me why the arrow in 4 denotes slow sinking (why are particles released from vertical migrators slower than those repackaged or from sloppy feeding?)
Answer: We agree with the remark and the arrow will be corrected from dotted arrow to solid arrow.

Table 1.- Caption should specify 'in fishes and squids' (as there are also luminescent bacteria in zooplankton, which are not shown here). "List of luminous bacterial species found in light organ symbiosis in fishes and squids"
Answer: We will add 'fishes and squids' to Table 1 caption as suggested.

| Species | Host Collection | Hosts | Light Organ Location |
|---|---|---|---|
| *Aliivibrio fischeri* (*Vibrio fischeri*) | *Euprymna* spp. Western Pacific (Fidopiastis et al., 1998) | **SEPIOLIDAE** *Euprymna* spp. *E. morsei* *E. berryi* *E. scolopes* *E. tasmanica* | E |
| | *Sepiola* spp. Mediterranean Sea, European Atlantic coast, Japan, Philippines (Fidopiastis et al., 1998) | *Sepiola* spp. *S. affinis* *S. atlantica* *S. intermedia* *S. ligulata* *S. robusta* | |
| | *Moconcentris japonica* Japan (Dunlap et al., 2007) | | |
| | *Cleidopus gloriamaris* East coast of Australia (Fitzgerald, 1977) | **MONOCENTRIDAE** *Monocentris* spp. *M. japonica* *Cleidopus* spp. *C. gloriamaris* | E |
| | *Caelorinchus* spp. Taiwan (*C. formosanus*) Japan (*C. multispinulosus*) (Dunlap et al., 2007) | **MACROURIDAE** *Caelorinchus* spp. *C. formosanus* *C. multispinulosus* | (I) |
| *Aliivibrio thorii* | *Sepiola affinis* Mediterranean Sea (Fidopiastis et al., 1998 ; Ast et al., 2007) | **SEPIOLIDAE** *Sepiola* spp. *S. affinis* | E |
| *Aliivibrio wodanis* * | *Sepiola* spp. Mediterranean Sea (Fidopiastis et al., 1998 ; Ast et al., 2007) | **SEPIOLIDAE** *Sepiola* spp. *S. affinis* *S. robusta* | E |
| *Photobacterium kishitanii* | *Opisthoproctus* spp. Atlantic Ocean (*O. grimaldii*) Atlantic Ocean and Indian Ocean (*O. soleatus*) (Haygood et al., 1992; Dunlap et al., 2007) | **OPISTHOPROCTIDAE** *Opisthoproctus* spp. *O. grimaldii* *O. soleatus* | (I) |
| | *Chlorophthalmus* spp. Japan (Dunlap et al., 2007) | **CHLOROPHTHALMIDAE** *Chlorophthalmus* spp. *C. acutifrons* *C. albatrossis* *C. nigromarginatus* | (I) |
| | *Caelorinchus* spp. Taiwan (*C. kishinouyei*) Japan (Other species) (Dunlap et al., 2007) | **MORIDAE** *Physiculus* spp. *P. japonicus* | I |
| | *Malacocephalus laevis* Indian Ocean (Dunlap et al., 2007) | **MACROURIDAE** *Caelorinchus* spp. *C. anatirostris* *C. denticulatus* *C. fasciatus* *C. hubbsi* *C. japonicus* *C. kamoharai* *C. kishinouyei* | (I) |
| | *Ventrifossa* spp. Japan (*V. garmani* and *V. longibardata*) Taiwan (*V. rhidodorsalis*) (Dunlap et al., 2007) | *Malacocephalus* spp. *M. laevis* *Ventrifossa* spp. *V. garmani* *V. longibarbata* *V. rhipidodorsalis* | |
| | *Physiculus japonicus* Japan (Dunlap et al., 2007) | | |
| | *Aulotrachichthys prosthemius* Japan (Ast and Dunlap, 2004) | **TRACHICHTHYIDAE** *Aulotrachichthys* spp. *A. prosthemius* | I |
| | *Acropoma hanedai* Taiwan (Kaeding et al., 2007; Dunlap et al., 2007) | **ACROPOMATIDAE** *Acropoma* spp. *A. hanedai* | I |

* firstly identified as *Vibrio logei* by Fidopiastis et al., 1998

| Species | Host Collection | Hosts | Light Organ Location |
|---|---|---|---|
| *Photobacterium leiognathi* | *Acropoma japonicum* Taiwan (Kaeding et al., 2007) | **ACROPOMATIDAE** *Acropoma spp.* *A. japonicum* | I |
| | *Gazza* spp. Philippines (Dunlap et al., 2004, 2007) | **LEIOGNATHIDAE** *Gazza* spp. G. achlamys G. minuta | |
| | *Leiognathus* spp. Taiwan (*L. equulus*) Okinawa (*L. fasciatus*) Philippines (*L. jonesi, L. philippinus*) Japan (*L. nuchalis*) Gulf of Siam (*L. splendens*) (Dunlap et al., 2004, 2007) | *Leiognathus* spp. L. equulus L. fasciatus L. jonesi L. nuchalis L. philippinus L. splendens | |
| | *Equulites* spp. Japan (*E. elongatus, E. rivulatus*) Philippines (*E. leucistus*) (Dunlap et al., 2004, 2007) | *Equulites* spp. E. elongatus E. leucistus E. rivulatus | I |
| | *Photopectoralis* spp. Japan (*P. bindus*) Philippines (*P. panayensis*) (Kaeding et al., 2007) | *Photopectoralis* spp. P. bindus P. panayensis | |
| | *Photolateralis* spp. Philippines (*P. stercorarius*) (Dunlap et al., 2007) | *Photolateralis* spp. P. stercorarius | |
| | *Secutor* spp. Philippines (Dunlap et al., 2007) | *Secutor* spp. S. insidiator S. megalolepis | |
| | *Uroteuthis noctilus* Sydney, Australia (Guerrero-Ferreira et al., 2013) | **LOLIGINIDAE** *Uroteuthis* spp. U. noctiluca | E |
| | *Rondeletiola minor* Mediterranean Sea, France (Guerrero-Ferreira et al., 2013) | **SEPIOLIDAE** *Rondeletiola* spp. R. minor *Sepiolina* spp. S. nipponensis | E |
| | *Sepiolina nipponensis* Japan (Nishiguchi and Nair, 2003) | | |
| *Photobacterium mandapamensis* | *Acropoma japonicum* Taiwan (Kaeding et al., 2007) | **ACROPOMATIDAE** *Acropoma spp.* A. japonicum | I |
| | *Gadella jordani* Taiwan (Kaeding et al., 2007) | **MORIDAE** *Gadella spp.* G. jordani | I |
| | *Photopectoralis* spp. Japan (*P. bindus*) Philippines (*P. panayensis*) (Kaeding et al., 2007) | **LEIOGNATHIDAE** *Photopectoralis spp.* P. bindus P. panayensis | I |
| | *Siphamia versicolor* Japan (Kaeding et al., 2007) | **APOGONIDAE** *Siphamia* spp. S. versicolor | I |
| *Vibrio harveyi* | *Uroteuthis chinensis* Thailand (Guerrero-Ferreira et al., 2013) | **LOLIGINIDAE** *Uroteuthis* spp. U. chinensis | E |
| | *Euprymna hyllbergi* Thailand (Guerrero-Ferreira et al., 2013) | **SEPIOLIDAE** *Euprymna* spp. E. hyllebergi | E |

| Species | Host Collection | Hosts | Light Organ Location |
|---|---|---|---|
| *Candidatus* Enterovibrio escacola | *Ceratias* spp.
NE Atlantic (*C.* sp)
Gulf of Mexico (*C. uranoscopus*)

*Lynophryne maderensis*
NE Atlantic

*Melanocetus johnsoni*
Gulf of Mexico and NE Atlantic

*Melanocestus murrayi*
Gulf of Mexico

*Chaenophryne* spp.
NE Atlantic

*Oneirodes* sp.
Gulf of Mexico

(Baker et al., 2019) | **CERATIIDAE**
  *Ceratias* spp.
    *C. uranoscopus*
    *C.* sp

**LINOPHRYNIDAE**
  *Linophryne* spp.
    *L. maderensis*

**MELANOCETIDAE**
  *Melanocetus* spp.
    *M. johnsoni*
    *M. murrayi*

**ONEIRODIDAE**
  *Chaenophryne* spp.
    *C. longiceps*
    *C.* sp
  *Oneirodes* spp.
    *O.* sp |  E |
| *Candidatus* Enterovibrio luxaltus | *Cryptopsaras couesii*
Gulf of Mexico and NE Atlantic
(Baker et al., 2019) | **CERATIIDAE**
  *Cryptopsaras* spp.
    *C. couesii* |  E |
| *Candidatus* Photodesnus blepharus | *Photoblepharon* spp.
Pacific Ocean (*P. palpebratus*)
Western Indian Ocean (*P. steinitzi*)
(Hendry and Dunlap, 2014) | **ANOMALOPIDAE**
  *Photoblepharon* spp.
    *P. palpebratus*
    *P. steinitzi* |  E |
| *Candidatus* Photodesnus katoptron | *Anomalops* spp.
Philippines
(Hendry and Dunlap, 2011) | **ANOMALOPIDAE**
  *Anomalops* spp.
    *A. katoptron* |  E |